# Role of Protein Phosphatases in Tumor Angiogenesis: Assessing PP1, PP2A, PP2B and PTPs Activity

**DOI:** 10.3390/ijms25136868

**Published:** 2024-06-22

**Authors:** Márton Fonódi, Lilla Nagy, Anita Boratkó

**Affiliations:** Department of Medical Chemistry, Faculty of Medicine, University of Debrecen, Egyetem tér 1, H-4032 Debrecen, Hungary; fonodi.marton@med.unideb.hu (M.F.); nagylilla@med.unideb.hu (L.N.)

**Keywords:** tumor angiogenesis, protein phosphatases, Ser/Thr phosphatases, Tyr phosphatases, endothelial cells, tumor cells, signaling pathways

## Abstract

Tumor angiogenesis, the formation of new blood vessels to support tumor growth and metastasis, is a complex process regulated by a multitude of signaling pathways. Dysregulation of signaling pathways involving protein kinases has been extensively studied, but the role of protein phosphatases in angiogenesis within the tumor microenvironment remains less explored. However, among angiogenic pathways, protein phosphatases play critical roles in modulating signaling cascades. This review provides a comprehensive overview of the involvement of protein phosphatases in tumor angiogenesis, highlighting their diverse functions and mechanisms of action. Protein phosphatases are key regulators of cellular signaling pathways by catalyzing the dephosphorylation of proteins, thereby modulating their activity and function. This review aims to assess the activity of the protein tyrosine phosphatases and serine/threonine phosphatases. These phosphatases exert their effects on angiogenic signaling pathways through various mechanisms, including direct dephosphorylation of angiogenic receptors and downstream signaling molecules. Moreover, protein phosphatases also crosstalk with other signaling pathways involved in angiogenesis, further emphasizing their significance in regulating tumor vascularization, including endothelial cell survival, sprouting, and vessel maturation. In conclusion, this review underscores the pivotal role of protein phosphatases in tumor angiogenesis and accentuate their potential as therapeutic targets for anti-angiogenic therapy in cancer.

## 1. Introduction

Tumors are characterized by rapid cell proliferation, heightened metabolic activity, and remarkable resilience, necessitating an increased supply of oxygen and nutrients compared to normal tissue cells. Initially, tumors exist in an avascular state, relying on the diffusion of oxygen and nutrients from the surrounding tissue. As the solid tumor expands beyond a defined/specified cell diameter, cells are no longer able to absorb either nutrients or oxygen. They create an altered microenvironment, characterized by hypoxia, ischemia, acidosis, and elevated interstitial pressure, that triggers the release of abundant growth factors and cytokines, thereby stimulating angiogenesis [1,2,3]. De novo synthesis of blood vessels in embryos by precursor cells, called angioblast or endothelial progenitor cells (EPCs), is referred to as vasculogenesis. These precursor cells differentiate into endothelial cells (ECs) and the proliferation of mesoderm-derived endothelial progenitor cells leads to the development of a primitive network of vessels, representing the initial signs of the vasculature [4,5] (Figure 1A). Pericytes and smooth muscle cells are recruited to stabilize the newly formed blood vessels. The surrounding extracellular matrix is remodeled and deposited to provide structural support to the blood vessels. The term “angiogenesis” originates from the Greek word “Angêion,” meaning vessel. It is defined as the formation of blood vessels from existing vasculature, therefore vasculogenesis is a crucial component of angiogenesis [6]. In addition to its pivotal role in embryogenesis, angiogenesis is also observed in adult organs, participating in physiological processes such as ovulation, pregnancy, wound healing, organ growth, cancer development, tumor cell proliferation, and the formation of metastasis [3]. Angiogenesis is divided into two main types, namely sprouting and intussusceptive angiogenesis. Sprouting angiogenesis occurs in hypoxic tissues where parenchymal cells, such as hepatocytes, neurons, and myocytes, respond to hypoxia by producing proangiogenic growth factors, predominantly vascular endothelial growth factor (VEGF) [7]. The process initiates with the destabilization of blood vessels, leading to capillary membrane degradation, pericyte detachment, and extracellular matrix weakening. VEGF stimulation induces a characteristic phenotype where tip and stalk cells are separated, and new vessel branches, called sprouts, are formed. Pericytes recover in the final step via the platelet-derived growth factor (PDGF)-B and angiopoietin (Ang)-1 signaling pathway, and molecular and adhesion junctions are formed among ECs, mainly through the Delta-Notch signaling pathway [7,8,9] (Figure 1B). Intussusceptive or splitting angiogenesis is more efficient and faster than sprouting angiogenesis. This process involves the rearrangement of existing ECs rather than cell proliferation or migration [10,11]. The first step of the splitting angiogenesis is the formation of interendothelial connection by the protrusion of opposite vessel walls using collagen bundles. As the endothelial bilayer perforates centrally, a transluminal pillar is forming that splits a single blood vessel into two. The pillar is enlarged by fibroblasts, myofibroblasts, and pericytes [10] (Figure 1C). Intussusceptive angiogenesis plays a crucial role in wound healing, capillary network growth, and environmental adaptation of tumor cells [8,10,12]. Tumor-associated vasculogenesis involves the recruitment and differentiation of endothelial progenitor cells to form new blood vessels within the tumor. This process involves the recruitment of endothelial progenitor cells from the bone marrow or resident cells within vessel walls, which undergo differentiation to form new blood vessels [13] (Figure 1D). Tumor cells exhibit a phenomenon known as “vascular mimicry”, where they behave like ECs by expressing endothelial-selective markers and anti-coagulant factors, forming vessels in the absence of vascular endothelium [14] (Figure 1E). The persistently released or upregulated various pro-angiogenic factors, activate ECs, pericytes, tumor-associated fibroblasts, endothelial progenitor cells, and immune cells, resulting in basement membrane disruption, extracellular matrix remodeling, and maintaining a highly active state of angiogenesis. Tumor cells can also integrate into the blood vessel wall, collaborating with pre-existing endothelium to form mosaic vessels [8,15].

Another way tumors ensure their supply is through vessel co-option, which refers to a process by which tumors use existing blood vessels from the surrounding normal tissue to facilitate their own growth and survival [14] (Figure 1F). Instead of inducing the formation of new blood vessels through angiogenesis, tumor cells exploit the pre-existing vasculature by adhering to and growing along the existing blood vessels. This allows the tumor to rapidly access oxygen and nutrients without the need for extensive angiogenic signaling or the development of a separate vascular network. Vessel co-option is often observed in cancers that grow in tissues with a rich vascular supply, such as the brain, liver, and lungs. In certain studies, tumor ECs have been noted to possess comparable somatic mutations to those found in the malignant cells of the tumor, suggesting a neoplastic source [16,17]. The trans-differentiation of cancer stem cells to ECs may occur within the tumor microenvironment and contributes to the formation of new blood vessels (Figure 1G).

Several proteins act as regulators of angiogenesis, either promoting (proangiogenic) or inhibiting (antiangiogenic) the process. The delicate balance between proangiogenic factors such as basic fibroblast growth factor (bFGF), VEGF, hepatocyte growth factor (HGF), and antiangiogenic factors like thrombospondin 1 (TSP-1), angiostatin, endostatin, integrins, or laminin β1 chain is crucial under physiological conditions [18]. These factors and their receptors belong to angiogenic signal transduction pathways, with their phosphorylation-dependent regulation playing a significant role [19]. When cells experience a lack of nutrients and oxygen, angiogenesis can be stimulated by upregulating activators or downregulating inhibitors, a factor that could be essential in tumor cell therapies [3]. In ECs, VEGF plays a critical role in developmental and post-natal angiogenesis by activating several cell signaling pathways [20].

## 2. Key Signaling Pathways Involved in Tumor Angiogenesis

Tumor angiogenesis is a complex and tightly regulated phenomenon. Various signaling pathways play critical roles in orchestrating the intricate molecular mechanisms underlying angiogenesis within the tumor microenvironment. Understanding these signaling pathways is essential for developing targeted therapeutic interventions to disrupt tumor angiogenesis and inhibit cancer progression. Crosstalk between these pathways coordinates cellular responses to angiogenic stimuli, amplifying pro-angiogenic signaling and promoting EC proliferation and migration. In this section, we will give a brief overview into the major signaling pathways involved in tumor angiogenesis (Figure 2).

### 2.1. Vascular Endothelial Growth Factor (VEGF) Pathway

VEGF proteins are produced by various cells, including tumor cells, immune cells, and stromal cells. Hypoxia, or low oxygen levels, is a major inducer of VEGF production in tumors. These VEGF proteins (VEGF-A, VEGF-B, VEGF-C, and VEGF-D) then bind their specific tyrosine kinase receptors (VEGFR), namely VEGFR-1 and VEGFR-2, found on the surface of ECs. VEGF-A/VEGFR-2 signaling plays a prominent role in orchestrating cellular responses associated with angiogenesis [21]. Once bound, VEGF triggers the activation of these receptors by autophosphorylation on Tyr residues, setting off a cascade of intracellular signaling events [22]. These pathways, which include Ras-mitogen-activated protein kinase (MAPK)/phosphatidylinositol 3′-kinase (PI3K)/protein kinase B (Akt), focal adhesion kinase (FAK), mammalian target of rapamycin (mTOR), and phospholipase Cγ (PLCγ)/protein kinase C (PKC), regulate key cellular processes such as proliferation, migration, survival, and permeability [22,23]. In response to VEGF signaling, ECs proliferate and migrate towards areas where new blood vessels are needed. They also gain increased resistance to apoptosis, ensuring their survival [24]. Moreover, VEGF-induced permeability changes facilitate the movement of fluids and cells across vessel walls. In cancer, dysregulated VEGF signaling can lead to excessive angiogenesis, fueling tumor growth, invasion, and metastasis [25]. Consequently, targeting the VEGF pathway has become a cornerstone of cancer therapy, with various anti-VEGF drugs approved for clinical use, such as the monoclonal antibody type Bevacizumab, Ramucirumab, and Ranibizumab or the recombinant fusion proteins like Aflibercept and Ziv-aflibercept.

### 2.2. Notch Signaling Pathway

The Notch pathway is initiated by interactions between transmembrane receptors (Notch receptors) and their ligands, which are also transmembrane proteins expressed on the surface of neighboring cells. There are four Notch receptors in mammals (Notch 1–4) and five ligands (Delta-like 1, 3, 4, and Jagged 1, 2) [26]. Delta-like ligand 4 (DLL4) has been demonstrated to serve as a pivotal ligand for the Notch pathway, stimulating angiogenesis [27,28]. Notch receptors are single-pass transmembrane proteins with extracellular, transmembrane, and intracellular domains. The extracellular domain contains multiple epidermal growth factor (EGF)-like repeats, while the intracellular domain contains RAM (RBP-Jκ-associated molecule) domain, ankyrin repeats, and a PEST sequence. Upon ligand binding, Notch receptors undergo a series of proteolytic cleavages, mediated by a disintegrin and metalloproteinase (ADAM) proteases and γ-secretase complex [29,30]. This results in the release of Notch intracellular domain (NICD), which translocates to the nucleus, where it forms a complex with C-promoter-binding factor (CSL) and Mastermind (MAM)/Lag-3 Mastermind-like (MAML) proteins [31]. This complex activates the transcription of Notch target genes, notably Hey/Hes proteins [32]. These proteins act as transcriptional repressors, regulating cell fate decisions, differentiation, proliferation, and apoptosis. A reciprocal interplay exists between the Notch and VEGF signaling cascades. VEGF can upregulate DLL4 expression, which in turn activates the Notch pathway to regulate angiogenesis [27]. Notch signaling can modulate VEGF receptor expression, affecting the responsiveness of ECs to VEGF stimulation [33]. The pathway’s dysregulation is associated with various diseases, including cancer, cardiovascular disorders, and neurodevelopmental defects. Aberrant Notch signaling can lead to uncontrolled cell proliferation and altered differentiation, contributing to tumor progression and other pathological conditions.

### 2.3. Fibroblast Growth Factor (FGF) Pathway

The FGF family comprises 22 members in humans, which are divided into several subfamilies based on structural and functional similarities. These growth factors interact with high-affinity tyrosine kinase receptors known as FGF receptors (FGFRs). FGFRs are transmembrane proteins with intracellular tyrosine kinase domains. There are four main FGFRs (FGFR1-4) in humans, each with multiple splice variants [34]. Binding of FGF ligands to FGFRs initiates receptor dimerization and autophosphorylation, leading to activation of downstream signaling pathways. Upon activation, FGFRs phosphorylate several intracellular signaling proteins, including FRS2 (FGF receptor substrate 2), which serves as an adaptor protein [35]. Phosphorylated FRS2 recruits and activates downstream effectors, including Ras-MAPK, phosphatidylinositol 3′-kinase (PI3K)-Akt, and PLCγ pathways. Activation of Ras leads to a cascade of phosphorylation events, ultimately resulting in the activation of MAPKs such as extracellular signal-regulated kinase 1/2 (ERK1/2). These kinases translocate to the nucleus and regulate gene expression involved in cell proliferation and differentiation. Activation of PI3K leads to the production of phosphatidylinositol (3,4,5)-trisphosphate (PIP3), which recruits Akt to the cell membrane. Akt is then phosphorylated and activated, leading to the regulation of various downstream targets involved in cell survival, growth, and metabolism. Activation of PLCγ leads to the hydrolysis of phosphatidylinositol 4,5-bisphosphate (PIP2) into inositol 1,4,5-trisphosphate (IP3) and diacylglycerol (DAG). IP3 triggers the release of calcium ions from intracellular stores, while DAG activates PKC, regulating cell proliferation and differentiation.

### 2.4. Platelet-Derived Growth Factor (PDGF) Pathway

The PDGF family consists of four ligands, PDGF-A, PDGF-B, PDGF-C, and PDGF-D, secreted as homodimers or heterodimers [36] with PDGF-BB having a predominant role in angiogenesis. The PDGF ligands bind to and activate two structurally related receptor tyrosine kinases known as PDGFR-α and PDGFR-β, of which PDGFR-β is specific to PDGF-BB. The expression of PDGFR-α is predominantly observed in cancerous cells, and its heightened levels are positively correlated with the progression of cancer [37,38]. These receptors are transmembrane proteins with intrinsic tyrosine kinase activity in their cytoplasmic domains. Upon ligand binding, PDGFRs dimerize and undergo autophosphorylation on specific tyrosine residues, which serves as docking sites for downstream signaling molecules in Ras-MAPK, PI3K-Akt, or PLC-γ pathway. PDGFR activation can also lead to the activation of Signal Transducer and Activator of Transcription (STAT) transcription factors, which regulate the expression of genes involved in cell proliferation, survival, and inflammation. PDGF signaling promotes pericyte recruitment and stabilization of newly formed blood vessels. Olaratumab (approved by FDA) is a monoclonal antibody that targets PDGFR-α. It is used as a treatment for advanced soft tissue sarcoma in combination with doxorubicin [39].

### 2.5. Integrin-Mediated Signaling

Integrins are heterodimeric receptors composed of α and β subunits. Integrins are involved in extracellular matrix (ECM) remodeling by binding to ECM proteins and facilitating the assembly and organization of the ECM. In angiogenesis, integrins primarily interact with ECM components such as fibronectin, collagen, laminin, and vitronectin, mediating cell adhesion, migration, and survival. Upon binding to ECM ligands, integrins initiate signaling cascades that lead to cytoskeletal rearrangements, focal adhesion assembly, and the formation of stress fibers, which are necessary for cell migration. Key signaling pathways activated by integrins in angiogenesis include FAK. FAK or protein tyrosine kinase 2 (PTK2) is a cytoplasmic tyrosine kinase that is activated upon integrin engagement [40]. FAK phosphorylation leads to the recruitment of adaptor proteins such as paxillin and the activation of downstream signaling pathways involved in cell migration and survival. Elevated level or activation of FAK have been observed in various types of human cancers [41] and its pivotal role in tumor progression was also shown [42]. Integrins can also regulate Rho GTPases such as Ras homolog family member A (RhoA), Ras-related C3 botulinum toxin substrate 1 (Rac1), and cell division control protein 42 homolog (Cdc42), which control cytoskeletal dynamics and cell migration and they are essential for the formation of lamellipodia, filopodia, and stress fibers, which are necessary for EC motility during angiogenesis [43]. Activation of ERK or PI3K/Akt pathway promotes EC proliferation and survival, contributing to angiogenic sprouting and blood vessel formation. Akt also enhances endothelial cell survival and inhibits apoptosis, allowing ECs to persist and form new blood vessels.

### 2.6. Angiopoietin-Tie Pathway

Angiopoietins (Ang) are vascular-specific growth factors consisting of Ang-1, Ang-2, Ang-3, and Ang-4. The transmembrane tyrosine kinase receptor (Tie) family comprises Tie-1 and Tie-2 receptors, with Tie-2 being the primary receptor involved in angiogenesis because of its regulatory effects on vascular stabilization [44,45,46]. Ang-1/Tie-2 can support opposing actions, angiostasis, and angiogenesis, as well as pro- and anti-tumor effects, depending on the cell type, environment, and the presence versus absence of endothelial cell-cell contacts [46,47]. Ang-1 binding induces cluster formation and auto-phosphorylation of Tie-2, creating a cross-phosphorylating complex on ECs and Ang-1 multimer-bound Tie-2 receptors on adjacent cells can assemble junctional signaling complex by connecting to each other [44,45]. In the presence of cell-cell contact Ang-1-Tie-2 link enhances and maintains cellular quiescence and vascular integrity by activating its main downstream targets PI3K and Akt, inducing cell survival and preventing endothelial apoptosis by inhibiting caspases and second mitochondria-derived activator of caspase (Smac) [44,45]. Furthermore, the PI3K/Akt signal indirectly activates β-catenin, subsequently inducing DLL4/Notch signaling [18]. Additionally, Ang-1/Tie-2 signaling strengthens inter-endothelial adhesion by enhancing junctional accumulation of vascular endothelial cadherin (VE-cadherin), stabilizing platelet endothelial cell adhesion molecule (PECAM)-1, inhibiting tumor necrosis factor-α (TNF-α) stimulated leukocyte transmigration, obstructing VEGF-induced permeability and inflammation, and inhibiting the inflammatory transcription factor nuclear factor kappa B (NFκB). All the above are required for maintaining of the integrity of non-proliferating endothelial monolayers during angiostasis in normal adult blood vessels [44,46,48,49,50]. In adults, the interruption of angiostasis is a common indicator of disease. The Ang-1/Tie-2/PI3K axis can induce secretion of plasmin and matrix metalloproteinase (MMP) -2, and it can provoke cytoskeletal changes enhancing the motility and migratory potential of cells [51,52]. Tie-2 can activate ERK1/2 thereby stimulating EC proliferation and enlargement of vascular structures [53]. EC proliferation and migration are regulated by VEGF through Akt activation and generation of reactive oxygen species (ROS). As a result of cell damage, ROS seem to play major role as mediator of angiogenesis during repair process. In cancer cells, a similar process can be observed as for ROS-induced angio- and neovasculogenesis, including tubule formation, cell migration, and proliferation [48]. Hypoxia, which evidently refers to the existence of tumor microenvironment, and inflammation enhance VEGF expression and cause angiogenic response [47,54]. Ang-1-induced angiogenesis seems to require endothelial-derived nitric oxide (NO), and Ang-1/Tie-2 signaling can stimulate cell motility and cytoskeletal remodeling through Downstream of kinase (Dok)–related protein (Dok-R)/Ras guanosine triphosphatases (GTPase)/non-catalytic region of tyrosine kinase adaptor protein (Nck) pathway [48,55]. Moreover, Ang-1 exposure of ECs enhances intracellular ROS, especially H_2_O_2_, which affects cell migration, tubule formation, and angiogenesis by modulating the signaling pathways of Akt and MAPK [48].

Ang-2 antagonizes the effects of Ang-1 by competitively binding to the Tie-2 receptor. Ang-2 binding obstructs the Ang-1-mediated phosphorylation, thereby suppressing Tie-2 signaling [45,51]. In quiescent ECs, Ang-2 is expressed to a small extent, and the pre-formed Ang-2 protein is stored in Weibel–Palade bodies. Ang-1/Tie-2/Akt axis-mediated FoxO1 (Forkhead box protein O1) inhibition keeps under control the production of Ang-2. During inflammation or hypoxia, including tumors, attenuated Ang-1/Tie-2 signaling unchains FoxO1, allowing Ang-2 production to be upregulated and rapidly released from Weibel–Palade bodies. High levels of Ang-2 provoke EC permeability and vessel regression, inducing hypoxia, which in turn enhances VEGF expression and inflicts an intense angiogenic response [44,45,46,47,50].

### 2.7. Transforming Growth Factor-Beta (TGF-β) Pathway

The TGF-β family comprises a group of multifunctional cytokines, including TGF-β1, TGF-β2, and TGF-β3 [56]. These cytokines exert their effects by binding to Ser/Thr kinase TGF-β receptors on the cell surface, existing as homodimers or heterodimers [57]. Upon binding of TGF-β ligands, the receptors undergo phosphorylation and activation, initiating downstream signaling cascades. The canonical TGF-β signaling pathway involves the activation of suppressor of mothers against decapentaplegic (Smad) proteins, which act as intracellular mediators of TGF-β signaling [56]. Upon receptor activation, Smad2 and Smad3 are phosphorylated and form complexes with Smad4. These complexes translocate to the nucleus, where they regulate the transcription of target genes involved in angiogenesis. In addition to the canonical pathway, TGF-β can also activate non-Smad signaling pathways, such as the MAPK pathway, PI3K/Akt pathway, and Rho GTPase signaling pathway [58]. These pathways regulate various cellular processes, including cell proliferation, migration, and differentiation, which are essential for angiogenesis. TGF-β interacts with other angiogenic factors, such as VEGF, FGF, and angiopoietins, to regulate angiogenesis. Crosstalk between TGF-β and these factors fine-tunes EC behavior and blood vessel formation during angiogenesis. Dysregulation of the TGF-β pathway is associated with various pathological conditions characterized by abnormal angiogenesis, including cancer, fibrosis, and vascular diseases. In cancer, TGF-β promotes tumor angiogenesis, invasion, and metastasis by modulating the tumor microenvironment and promoting the epithelial-mesenchymal transition (EMT) of cancer cells. Given its critical role in angiogenesis and disease pathogenesis, the TGF-β pathway represents a potential therapeutic target for angiogenesis-related disorders [59]. Strategies aimed at modulating TGF-β signaling, such as TGF-β receptor inhibitors or Smad pathway inhibitors, hold promise for the treatment of cancer, fibrosis, and other diseases characterized by dysregulated angiogenesis.

## 3. Role of the Main Ser/Thr Phosphatases in Tumor Angiogenesis

Reversible protein phosphorylation is one of the most common modifications of proteins to regulate their activity, conformation, interacting partners, or cellular fate. Protein kinases attach a phosphate group to Ser, Thr, or Tyr side chain of proteins [60], while protein phosphatases are able to remove it by hydrolysis [61,62]. Around 90% of phosphorylation proceeds on Ser/Thr chains and only 2% occurs on Tyr side chains [63,64,65]. Protein phosphatases are classified as phospho-Ser/Thr or phospho-Tyr phosphatases and there are some dual specificity enzymes that can dephosphorylate all the three amino acid sidechains. The current classification of phospho-Ser/Thr protein phosphatases (PSTPs) depends on the primary structure of catalytic subunit [66,67]. Phosphoprotein phosphatases (PPPs) and metal-dependent protein phosphatases (PPMs) represents the two large families, while the third, smallest family is called the small CTD phosphatases (FCP/SCP) subfamily [65,67]. PPP family has been originally divided into three subfamilies, PP1, PP2A, and PP2B; later, some novel members (PP4, PP5, PP6, PP7) were added to the family [66,67,68,69]. The holoenzyme consists of one or more regulatory subunits that can bind to the N-or C-terminal of the catalytic domain [60,65]. Regulatory subunits play critical roles in fine-tuning cellular signaling pathways by regulating the activity and localization of protein phosphatases. Their ability to target substrates, modulate enzymatic activity, and dictate subcellular localization allows for precise and dynamic control of protein phosphorylation events, influencing various cellular processes. This chapter summarizes the pivotal roles of PPPs in tumor angiogenesis signaling, highlighting their regulatory mechanisms.

### 3.1. Protein Phosphatase 1 (PP1)

PP1 is a putative Ser/Thr phosphatase functioning as a heterocomplex, consisting of a catalytic bound to one of the numerous regulatory subunits also known as regulatory interactors of protein phosphatase one (RIPPOs). The catalytic subunit of PP1 (PP1c) is encoded by three genes, resulting in three primary isoforms: PP1α, PP1β/δ, and PP1γ [65,70]. More than 200 interacting partners of PP1c have been identified to date, emphasizing the enormous range of cellular processes the enzyme regulates [71]. Changes in frequency of genomic alterations as well as in expression of PP1 in tumors proves valuable for tumor progression and are used as biomarkers for many tumor types [72]. PP1 plays an extensive role in tumorigenesis and tumor angiogenesis, tumor microenvironment, and metastatic cascade (Figure 3). The emerging role of PP1cα, PP1cγ, and interestingly PP4c was reported, as knock down of these genes resulted in vascular perfusion without effecting EC migration [73]. PP1β has been found to regulate EC migration in primary human umbilical vein endothelial cells (HUVECs) [74]. In a hypoxic environment, PP1β showed increased levels in vitro. PP1β knockdown induced cytoskeletal reorganization, loss of focal adhesion sites, and impairment of FAK activation. FAK mediates the turnover of focal adhesions and EC migration through the formation of stress fibers [74,75]. PPP1CA gene amplification was shown in prostate cancer cells, resulting in cell invasiveness via the regulation of MAPK signaling pathway [76]. Upregulation of PP1γ isoform and its involvement in NF-κB signaling cascade through dephosphorylation of p65 in human glioma was also shown [77]. TGF-β interplays with PP1 through regulating paxillin expression. Upregulation of paxillin by TGF-β was blocked by inhibiting PP1 activity, indicating that paxillin is a substrate of PP1 [78]. The phosphorylation state of paxillin affects its interaction with actin filaments, which are necessary for cell motility [78].

Many PP1c regulatory subunits have the typical RVXF PP1c binding motif, such as tensin1, a multifunctional scaffold protein. Mutation of the binding motif (F302A) of tensin1 prevents its interaction with PP1cα and increases the migration and invasion of human breast cancer cells [79]. The role of PP1 regulatory subunit 1A (PPP1R1A), also called inhibitor-1, has been shown in tumor development. It was found that in Ewing sarcoma, inhibitor-1 is upregulated, and it is essential in oncogenic transformation and tumor formation by regulating cell migration and metastasis [80]. Protein kinase A (PKA) mediated phosphorylation of inhibitor-1 increase its binding and inhibitory effect on PP1, resulting in an increased phosphorylation of many PP1 substrates that play an important role in the proliferation of Ewing’s sarcoma [80].

Myosin phosphatase-targeting subunit (MYPT) family members are one of the well-studied PP1 regulatory subunits [81]. MYPT1 has been shown to inhibit EC capillary tube formation and tumor angiogenesis in mouse models using in vivo tumor angiogenesis assay. In prostate tumors, microRNA (miR)-30d overexpression has been associated with critical prognosis. miR-30d promotes tumor angiogenesis by inhibiting MYPT1, resulting in an increased phosphorylation level of c-Jun and an increased expression of VEGF-A protein [82,83]. The RhoA/Rho Associated coiled-coil containing protein kinase (ROCK) pathway plays a crucial role in angiogenesis by regulating cell motility, permeability, proliferation, and migration [84]. MYPT1 is phosphorylated by ROCK on Thr696 and Thr853 side chains [85]. Inhibition of ROCK by Wf-536, a pyridine derivative type ROCK inhibitor or by Fasudil, reduces phosphorylation level of MYPT1 and therefore regulates myosin phosphatase and eventually the motility of EC and prostate cancer cells [86,87]. Calpeptin activation of ROCK increased the phosphorylation level of MYPT1 and decreased tube formation of pulmonary ECs [88]. PI3Kα is necessary for cell remodeling and inhibits the actomyosin contractility via NUAK family kinase 1 (NUAK1)/MYPT1/myosin-light-chain phosphatase (MLCP) pathway [89]. NUAK1 phosphorylates MYPT1 on Ser445 and its blockade is able to restore the endothelial phenotype caused by PI3K inactivation [90]. Several other studies showed the involvement of MYPT1 in cell migration and adhesion [91,92,93].

TGF-β inhibited membrane associated protein (TIMAP) is another member of the MYPT family that has also been shown to regulate angiogenesis via its interacting partners or substrate proteins. TIMAP is downregulated by TGF-β treatment in EC and regulates PP1c activity in a phosphorylation dependent manner [94,95,96]. In ECs, TIMAP-PP1c complex regulates the phosphorylation of the non-integrin laminin receptor 1 (LAMR1) [97,98]. LAMR1 upregulation has been shown in injured microglia and in vasculopathic neovascularization [99]. It is also required for in vivo and in vitro angiogenesis [100,101]. Obeidat et al. showed that inhibition of TIMAP expression in glomerular ECs decreased cell proliferation and angiogenic sprout formation [102]. Controversially, TIMAP silencing had no effect on cell proliferation, but increased in vitro tube formation, cell migration, and caused angiogenic phenotype in pulmonary artery ECs [103]. Endothelin converting enzyme-1 (ECE-1), a metalloprotease responsible for producing biologically active endothelin-1 (ET-1), has been identified as a substrate of TIMAP-PP1c [103]. ECE-1 is under the phosphorylation control of PKC and the TIMAP-PP1c complex, regulating its activity and subcellular localization. Silencing of TIMAP in pulmonary ECs resulted in increased ECE-1 activity and increased ET-1 secretion. The enhanced ET-1 secretion altered angiogenic properties of ECs such as actin-reorganization and accelerated in vitro tube formation and cell migration. Increased ET-1 secretion has been shown to direct ECs towards an angiogenic phenotype as well as regulating sprouting of new vessels in the tumor microenvironment [103]. TIMAP also promotes angiogenesis through phosphatase and tensin homolog (PTEN) mediated Akt inhibition [102]. VEGF signaling results, among other things, in the activation of the PI3K-Akt pathway [104]. PIP3 recruits Akt to the cell membrane resulting in Akt phosphorylation and activation by phosphoinositide-dependent kinase (PDK1) on Thr308 and mTOR complex 2 on Ser473 [105]. Activated Akt regulates a plethora of cellular processes such as angiogenesis, cell migration, proliferation and survival through phosphorylation of a number of downstream proteins on Ser/Thr residues [106]. PTEN suppresses the activation of Akt by dephosphorylation of PI3K. TIMAP regulates PTEN phosphorylation on Ser370, thus repressing PTEN activity, which, in turn, favors the activation of Akt. TIMAP depletion in glomerular ECs inhibited the formation of angiogenic sprouts [102].

Moesin-ezrin-radixin like protein (merlin) or neurofibromin 2 is a tumor suppressor of the ezrin-radixin-moesin (ERM) family regulating cell junctions and is involved in the Hippo signaling pathway [107,108]. PP1c-MYPT1 and PP1c-TIMAP both have been shown to dephosphorylate merlin on Ser518. The phosphorylation state of merlin determines its subcellular localization and activity in ECs as phospho-merlin promotes cell migration [109]. Annexin A2 (ANXA2) play a pivotal role in angiogenesis and tumor angiogenesis (reviewed in [110]). ANXA2 can be phosphorylated on Ser11, Thr23, and Ser25, regulating its activity and subcellular localization [111]. Phoshpo-Ser25 ANXA2 is a substrate of TIMAP-PP1c [112]. Nonphosphorylated ANXA2 shows diffuse localization while phospo-Ser25 ANXA2 showed a distinct membrane localization and phosphorylation disrupted S100A10 interaction, a key interacting partner of ANXA2 [112]. Myosin-dependent tip and stalk cell cytoskeletal reorganization is a critical step in angiogenesis. Myosin phosphatase, consisting of PP1β and MYPT1, regulate the dephosphorylation of myosin light chain (MLC) thus facilitating part of the cross-bridge cycling [113]. TIMAP competes with MYPT1 for PP1β, blocking the active site of PP1 and inhibiting its myosin phosphatase activity [114].

The phosphatase actin regulator (PHACTR) family members have a PP1c binding domain in their structure [115]. PHACTR-1 has been identified as a highly conserved protein expressed mainly in the brain as a PP1 and actin binding protein regulating PP1 activity and F-actin remodeling [115]. Jarray et al. reported VEGF-A165 dependent expression of PHACTR-1 in HUVECs, implying a possible role in angiogenesis [116]. PHACTR-1 depletion led to considerable decrease in tube formation and stability by activation of death receptors triggering apoptosis. PP1 is known to be a crucial regulator of apoptosis and cell cycle with well-studied dysregulation in numerous cancer types [72,117]. PHACTR-1 depletion also reduced PP1α activity, which, in turn, disrupted VEGF-A165 induced actin remodeling and lamellipodia dynamics, showing the great importance of the VEGF-PHACTR-1- PP1 signaling in angiogenesis [19].

Recently, FoxO1 has been shown to regulate angiogenesis by affecting MLC2 phosphorylation [118]. FoxO1 targets PP1 regulatory inhibitor subunit 14C (PPP1R14C), an inhibitor of PP1, regulating PP1-MLCP. FoxO1 was found to increases PPP1R14C gene expression, thus inhibiting MLCP, which in turn increases MLC2 phosphorylation. Increased MLC2 phosphorylation promoted cell elongation during angiogenesis.

Profilin-1 (Pfn-1) is an actin binding protein that plays a role in ECs motility and capillary morphogenesis [119]. Pfn-1 is a multi-faceted protein regulating processes such as cell motility mainly through regulation of actin polymerization [120]. VEGFR-2 activation induces Pfn-1 phosphorylation on Tyr129 by Src, promoting actin polymerization in the cell leading edge during angiogenesis [121]. Phosphorylation on Tyr129 increased endothelial angiocrine factor secretion and induced hypoxia-induced factor-1α (HIF-1α) stabilization. Pfn-1 is also dephosphorylated on Ser137 by PP1 [122] and overexpression of Pfn-1 phosphonull mutant showed decreased migration, growth, and invasion on tumorigenesis in breast cancer cells, showing the promoting effect of Pfn Ser137 phosphorylation [123].

Research on PP1 modulators in cancer therapy is still in the early stages, and clinical trials involving these modulators are limited. Current research primarily focuses on understanding the mechanistic roles of PP1 and developing selective inhibitors or activators. For instance, ongoing studies are exploring PP1 as a therapeutic target by designing small molecules that can selectively inhibit or activate its function. However, translating these findings into clinical trials has been slow due to the need for high specificity and the risk of adverse effects. As PP1 is always associated with one of its regulatory subunits and does not exist in its free catalytic form in cells, targeting of RIPPOs are more promising [72].

### 3.2. Protein Phosphatase 2A (PP2A)

PP2A is a ubiquitously expressed Ser/Thr phosphatase that regulates a vast variety of cellular processes [124]. The core enzyme complex consists of a heterodimer comprising a scaffolding A subunit and the catalytic C subunit [61,65,124]. When the heterodimer complex associates with the regulatory B subunit, a heterotrimeric PP2A is formed [65,124,125]. The two isoforms of the scaffolding subunit PP2A Aα and -Aβ are encoded by PPP2R1A and PPP2R1B, respectively. While the isoforms share 86% identity in amino acid sequence, PP2A Aα is dominantly present in mammalian cells [126]. The catalytic subunit (PP2A C) has two isoforms encoded by two different genes, PPP2CA and PPP2CB, that are 97% identical in amino acid sequence [127]. The very variable regulatory B subunit is divided into four subfamilies, B (PR55), B’ (PR56), B’’ (PR72), and B’’’ (Striatin or STRN), each containing several isoforms and splicing variants [65,128]. There are little-to-no similarities between the subfamilies, resulting in numerous forms of the heterotrimeric PP2A holoenzyme, thereby ensuring a wide range of potential substrates and providing regulatory function in tissue specificity, cellular localization, and cellular function [129]. The ever-increasing number of regulatory subunits of PP2A provides immense range of substrate specificity throughout several signaling pathways (Figure 4). PP2A is often considered a tumor suppressor because of its important role in the dephosphorylation of several oncoproteins such as Akt, ERK, c-myc, or ribosomal protein S6 Kinase β-1 (p70S6K) [130]. PP2A is involved in key tumor angiogenesis-related processes, such as cell migration, proliferation, apoptosis, and the underlying signaling pathways mediating them. Abnormal PP2A expression and activity are often associated with tumorigenesis, cancer progression, and many diseases [131]. Mutation of PPP2R1A gene creates a dysfunctional PP2A enzyme that corresponds to a reduced expression influencing the enzymatic activity of the PP2A holoenzyme [132]. This mutation is frequently found in cancer cells and promotes the phosphorylation of specific kinases that participate in angiogenesis. Downregulation of PPP2CA gene promotes an aggressive phenotype of prostate cancer cells, including migration and invasion, while overexpression of PP2A C increases the expression of epithelial markers of EMT [133].

The therapeutic potential of indirectly or directly targeting PP2A with pharmaceutical agents has long been recognized; however, reviewing the processes regulated by PP2A presents a formidable challenge, given its extensive involvement across various cellular pathways [134]. LB-100, also known as CCI-779, is a potent inhibitor of PP2A and has been investigated for its potential therapeutic applications [135]. The first in-human phase I study started in 2013 at the Mayo Clinic and was conducted to evaluate the safety, tolerability, and preliminary antitumor activity of LB-100 in patients with advanced solid tumors [136]. Inhibition of PP2A in pancreatic cancer by LB-100 synergistically enhances the activity of doxorubicin, an anticancer agent, inhibiting cancer growth [137]. In vivo xenograft models showed increased expression of HIF-1α and VEGF due to LB-100 treatment, resulting in increased microvessel density and blood perfusion. The authors showcase LB-100′s possible role as a chemosensitizer, increasing the efficacy of antitumor pharmaceutical agents. In mucoepidermoid carcinoma, LB-100 treatment also sensitized cancer cells to cisplatin treatment. In response to insulin, LB-100 treated cells exhibit increased phosphorylation of insulin receptor substrate-1 (IRS-1) on serine residues, decreased expression of PI3K p110 alpha subunit, as well as lower phosphorylation levels of Akt on Ser473 and Thr308, a known regulatory pathway of cell invasion, angiogenesis, and metastasis [138].

Prolyl hydroxylase domain-containing protein 2 (PHD2) is the most important isozyme regulating the stability of HIF-1. PHD2 is subjected to phosphorylation on Ser125 by a downstream target of mTOR, namely p70S6K kinase, while PP2A B55α holoenzyme dephosphorylates PHD2. During hypoxia, inhibition of mTOR blocks p70S6K activity, promoting dephosphorylation and inactivation of PHD2, which leads to HIF-1α accumulation and stabilization [139]. Tang et al. showed that the PP2A B56α causes inhibition of cell proliferation through polycystin-1 (PC-1) in HEK293T cells by regulating dephosphorylation of downstream effectors of mTOR [140]. The scaffold protein autophagy and beclin 1 regulator 1 (AMBRA1) is a downstream target of mTOR and promotes the interaction between c-myc and PP2A. Inhibition of mTOR increases PP2A activity towards Ser62-phosphorylated c-myc, resulting in the dephosphorylation and destabilization of c-myc and a reduced cell division rate [141].

PP2A regulates the activity and subcellular localization of histone deacetylases 7 (HDAC7) and other members of the class IIa HDACs through dephosphorylation. HDAC7 plays a crucial role in regulating vascular integrity primarily by repressing MMP-10 [142]. Inactivation of PP2A leads to hyperphosphorylation of HDAC7 and subsequent nuclear export. Notably, the 14-3-3 protein, an interacting partner of HDACs, shares the same binding site on serin residues with PP2A dephosphorylation sites, thereby masking these phosphorylation sites and preventing dephosphorylation by PP2A. Inhibition or knockdown of PP2A results in a significant increase in MMP-10 expression and a considerable reduction in tube formation in vitro [143]. The same research group later identified Bα regulatory subunit of PP2A as the primary factor in this interaction, required for vessel lumen stabilization and angiogenesis. In PP2ABα/HDAC7/ArgBP2 signaling, ArgBP2 (sorbin and SH3 domain-containing protein 2) was found to be the key effector regulating EC cytoskeletal properties by modulating RhoA activity, a crucial player in cytoskeletal dynamics [144]. Additionally, HDAC7 undergoes dephosphorylation by the myosin phosphatase MYPT1-PP1β complex, although this interaction does not dependent on 14-3-3 protein binding, suggesting alternative phosphorylation sites compared to those targeted by PP2A dephosphorylation [145]. HDACs are mediators of the VEGF signaling cascade; VEGF binding activates protein kinase D (PKD), which phosphorylates HDAC7, thereby facilitating the expression of angiogenesis-related genes such as regulator of calcineurin 2 (RCAN2) and nuclear receptor Nur77 [146].

Endostatin, a proteolytic fragment of collagen XVIII, is a well-known inhibitor of angiogenesis [147]. It induces dephosphorylation of ERK1/2 kinase by regulating PP2A [148]. In vivo experiments have demonstrated that endostatin-mediated PP2A activation and subsequent ERK1/2 kinase dephosphorylation lead to the retraction of ECs and newly formed vessels during vascular morphogenesis.

More and more evidence is emerging suggesting the intricate interplay between neuronal-derived angiogenic signals and tumor angiogenesis and cancer microenvironment [149]. Netrin-1 was originally described as a secreted protein regulating axon guidance; however, its role is far more multifaceted. Netrin-1 promotes angiogenesis by regulating cell survival of ECs via its interaction with the unc-5 netrin receptor B (UNC5B) [150]. The unbound UNC5B receptor activates Death-Associated-Protein kinase (DAPK), which in turn leads to caspase 3 activation as subsequent apoptosis. Netrin-1 binds UNC5B, thus blocking the proapoptotic effect of the receptor and its downstream elements [150]. The PP2A Aβ, which has been linked to cancer development [128], has been shown to associate with the UNC5B/DAPK complex, leading to dephosphorylation of DAPK on Ser308 and consequent activation [151,152,153]. In the case of Netrin-1 binding, the cellular inhibitor of protein phosphatase 2A (CIP2A) is recruited to the complex, which represses PP2A activity. CIP2A also interacts with Akt, regulating tube formation through glycogen synthase kinase-3 (GSK-3)-β/β-catenin pathway [154]. Inhibition of PP2A by CIP2A prevents dephosphorylation of oncoproteins involved in proliferation of cancer cells and tumor growth. Euxanthone, a naturally occurring organic compound, has an anti-cancer effect that functions by inhibiting EMT and reducing metastatic potential through repression of CIP2A and regulation of PP2A activity. FTY720, an activator of PP2A, reduces the level of CIP2A, implying the consideration of PP2A as a target in cancer therapy [155]. Although CIP2A inhibitors are not as advanced in clinical trials as LB100, their potential in cancer therapy through PP2A modulation is being actively researched. These inhibitors might indirectly affect angiogenesis by restoring PP2A’s regulatory functions [156].

Endothelial nitric oxide synthase (eNOS) is a key regulator of numerous cellular mechanisms, including angiogenesis [157]. The complex regulation of eNOS activity is mediated by a multitude of posttranslational modifications such as S-nitrosylation, acetylation, glutathionylation, O-glycosylation, and phosphorylation (reviewed in [158]). Regulatory Ser/Thr phosphorylation sites include human Ser114, Thr495, Ser615, Ser633, and Ser1177. PP2A is involved in the dephosphorylation of Thr495, which, in its phosphorylated form, attenuates eNOS activity, and Ser1177, which is the main activating modification in its phosphorylated form. Individual phosphorylation sites are also capable of influencing each other, adding another layer of complexity of regulation of the enzyme [159]. Netrin-1 stimulates angiogenesis through a deleted in colorectal cancer (DCC)-dependent ERK-1/2-eNOS feed-forward mechanism [160]. Upon Netrin-1 treatment of bovine aortic endothelial cells (BAEC), increased phosphorylation of Ser1179 (human Ser1177), Ser116 (human Ser114), and dephosphorylation on Thr497 (human Thr495), three known regulatory phosphorylation sites of eNOS, were detected. Increased eNOS activity led to elevated bioavailable nitric oxide radical (NO^·^) production, promoting endothelial angiogenic properties such as cell proliferation and migration. PP2A dephosphorylates eNOS on Ser1179 and Thr497, but not on Ser116. In BAEC, the B56α regulatory subunit-containing PP2A holoenzyme has been identified to dephosphorylate eNOS of Ser1179 in response to all-trans retinoic acid (ATRA) treatment [161], decreasing NO release. PP2A also regulates eNOS gene expression through Sp1 transcriptional factor dephosphorylation [162]. NO is known to regulate TSP-1 expression in a dose-dependent manner via the ERK and MAPK pathways [163]. TSP-1 is multi-domain matricellular protein known be a master regulator of angiogenesis [164]. We recently demonstrated that TSP-1 is a direct substate for the B55α subunit containing PP2A holoenzyme, which regulates angiogenic properties and processes of ECs [165]. Our findings revealed that TSP-1 is phosphorylated on Ser93 by PKC and dephosphorylated by PP2A. Silencing of B55α resulted in decreased TSP-1 expression, as well as alterations in wound closure, tube formation, and disruptions in spheroid formation in vitro. B55α depletion or inhibition of PP2A induces apoptosis, thereby delaying tumor progression in immature blood vessels but not in mature and stabilized ones, suggesting that vascular remodeling is mainly regulated by B55α [166]. PP2A-B55 α also regulates angiogenic properties through its interaction with flotillin-1. Dephosphorylation of flottilin-1 on Ser315 by PP2A affects its subcellular localization and promotes EC migration and tube formation [167].

The Hippo pathway was initially described as the fundamental signaling in developmental biology; however, since then, it has been associated with a wide range of processes such as tumorigenesis, cancer progression and metastasis, and tumor angiogenesis [108,168]. At the core of the Hippo pathway is a phosphorylation cascade event where mammalian sterile 20-like kinase 1/2 (MST1/2), the human ortholog of Hippo protein, in complex with Salvador Family WW Domain Containing Protein 1 (SAV1), phosphorylates and activates Large Tumor Suppressor Kinase 1/2 (LATS1/2), which in turn phosphorylates the main effectors Yes-associated protein (YAP)/WW-domain-containing transcription regulator 1 (TAZ). Dephosphorylated YAP/TAZ is present in the nucleus and acts as a transcription co-activator in concert with their interacting partners transcriptional enhancer factor (TEA)-domain family (TEAD1-4). When phosphorylated, YAP/TAZ is either sequestered by 14-3-3 proteins for activity in the cytoplasm or primed for ubiquitin-mediated protein degradation. Key upstream elements in connection with angiogenesis include: (A) Growth factors and receptors such as VEGF, Angiopoetin-2, epidermal growth factor receptor (EGFR), G protein-coupled receptors (GPCRs), TGFR, and Wingless/Integrated (Wnt) signaling; (B) Cell polarity and adhesion components: Crumbs homolog 1 (CRB1), Kibra, merlin, cadherins, as well as members of the angiomotin family (angiomotin-like 1, angiomotin-like 2); (C) Mechanical forces affecting cell geometry. An important regulator of Hippo signal transduction is the STRN-interacting phosphatase and kinase (STRIPAK) complex [169]. The STRIPAK complex consist of the PP2A AC dimer with a STRN-family regulatory subunit (B”’) acting as a scaffold for the complex [170,171], several kinases and different scaffolding proteins such as calmodulin (CaM) [172], Mob3, the germinal center kinase III (GCKIII) such as Serine/Threonine Kinase (STK) 24, STK25, MST4, misshapen like kinase 1 (MINK1) [173], family with sequence similarity 40 (FAM40) proteins, and cerebral cavernous malformations 3 protein (CCM3) [174]. PP2A in the STRIPAK complex can dephosphorylate MST1/2 kinases, thus activating the YAP/TAZ [175]. SAV-1 also regulates the PP2A-STRIPAK complex by inhibiting PP2A phosphatase activity through the adaptor protein SLMAP [176]. A recent study showcased that PP2A-B55α is capable of inhibiting LATS1 activation outside of the STRIPAK complex and showed evidence of direct interaction between PP2A-B55α and YAP, an essential interaction for YAP-promoted gene transcription in pancreatic cancer cells [177]. PP2A mediated YAP hypophosphorylation was also observed in endothelial cells, and inhibition of PP2A decreased VEGF-mediated angiogenesis [178]. The STRIPAK complex plays crucial role in cell migration of mammalian cells; furthermore, it is an important component in actin cytoskeleton and focal adhesion formation [179]. In the STRIPAK complex, proteins like FAM40A/STRIP1 or FAM40B/STRIP2 ensure the specificity of PP2A/STRN phosphatase. Loss of scaffold proteins results in decomposition of the STRIPAK complex and causes hyperphosphorylation of the PP2A-STRN target proteins [170]. STRIP1, STRIP2, and STRN3 have been identified as regulators of actomyosin, while MST3 and MST4, members of GCKIII kinases, regulate cancer cells migration [180]. Loss of STRIP1 inhibits the cell cycle and proliferation in breast tumor cells by induction of p21 and p27 expression [181]. Many members of the STRIPAK complex have been directly linked to angiogenic-related processes. STRIP1 and STRIP2 depletion led to an increase of stress fibers and impaired angiogenesis of endothelial cells in vitro [182]. In cancer cells, STRIP1, STRIP2, and STRN3 regulate cell migration and metastasis by negatively influencing MST3 and MST4 kinases, which act through the ezrin-radixin-moesin family proteins to promote the actomyosin machinery by phosphorylating PP1 and PPP1R14A-D [182]. Depletion of STRIP2 increases cells migration of MD Anderson-Metastatic Breast-231 cells [180] and reduces proliferation, invasion, and migration of non-small-cell lung cancer cells, while overexpression of STRIP2 in human lung cancer cells causes enhanced proliferation, invasion, and migration [183]. STRIP2 contributes to cell proliferation and migration through the p38-Akt-MMP-2 pathway in mouse aortic smooth muscle cells and inhibits the expression of VEGF [184].

CCM3 has also been reported to regulate VEGFR-2 endocytosis and degradation [185]. Expression changes of PP2A and STRN genes have also been reported as a prognostic marker, as well as influenced chemo-sensitivity in breast cancer [186]. CCM3 is known to affect endothelial angiogenic properties by regulation of the Notch, VEGFR, and ERK signaling pathways [187]. In glioblastomas, CCM3 downregulation resulted in higher microvascular density [188]. STRN4 binds and inhibits MAP4Ks, thus regulating the Hippo pathway by modulation of the MAP4K/LATS1/2 kinase cascade [189]. Striatin also regulates cell junctions in ECs by interacting with several adherent junction and tight junction proteins such as occludin, adenomatous polyposis coli (APC), and E-cadherin [190].

c-Jun N-terminal kinase (JNK) participates in the regulation of mesenchymal and perivascular cells response to angiogenic signals in injured tissues and tumors. Furthermore, it regulates the TGF-β induced proangiogenic responses [191]. PP2A inhibitors can activate the JNK pathway through the block of G2/M cell cycle induction and the inhibition of tumor cell growth [192]. RhoB small GTPase, a tumor suppressor, blocks tumor angiogenesis by inhibiting cell proliferation and invasion of cancer cells. Moreover, it promotes Akt1 dephosphorylation by PP2A, thereby inducing apoptosis of osteosarcoma and blocks metastasis [193,194].

NF-κβ transcription factor has been described to be involved in angiogenesis, and an inhibitor of nuclear factor kappa-B kinase subunit beta (IKKβ) has been shown to participate in the formation of tumor angiogenesis [195]. PP2A is involved in the regulation of NF-κβ pathway via the B subunit specificity of the holoenzyme. The PP2A AC dimer and PP2AB’”C trimer can dephosphorylate IKKβ and p65/RelA (subunit of NF-κβ), whereas IκBα is subjected to dephosphorylation by PP2A AC dimer, PP2ABC, and PP2AB’”C trimers. These results suggest that PP2A regulates NF-κβ activity and IKKβ independently [196].

Studying PP2A activators and inhibitors in clinical trials is challenging due to the enzyme’s involvement in multiple complex signaling pathways, making it difficult to predict and control the broad spectrum of cellular responses. The impact on downstream signaling can vary depending on which PP2A holoenzyme combination is modulated. The need for precise targeting of specific PP2A holoenzymes to achieve desired therapeutic outcomes without off-target effects adds another layer of complexity to the development and testing of these agents.

### 3.3. Calcineurin or PP2B

Calcineurin (also called protein phosphatase 2B, CaN, PPP3) is highly conserved Ca^2+^/calmodulin-dependent Ser/Thr phosphatase belonging to the family of PPP. CaN is a constitutive heterodimer composed of two subunits: a 58–64 kDa catalytic subunit, CnA, and a 19 kDa regulatory subunit, CnB. The isoforms α, β, and γ of CnA are encoded by PPP3CA, PPP3CB, and PPP3CC, respectively. CnAα is mainly expressed in neurons, while CnAß is ubiquitous and CnAγ is predominantly present in testes [197]. The calmodulin binding subunit CnA consist of a globular catalytic domain, an α-helical region that binds to CnB, a regulatory domain, and an autoinhibitory domain [198]. CnA utilizes two cofactors—Zn^2+^ and Fe^2+^—that are required for its catalytic activity. The small regulatory subunit CnB shows high structural similarity to calmodulin and comprises four EF-hand domains, each of which binds to four Ca^2+^ ions arranged in pairs within each lobe. One pair displays low affinity for Ca^2+^ ions while the other exhibits high affinity binding [198,199]. CnB1 and CnB2 are encoded by PPP3R1 and PPP3R2, respectively. CnB1 is ubiquitously expressed and forms heterodimers with CnAα or CnAß while CnB2 is associated with CnAα in testes. CaN is inactivated at resting Ca^2+^ ion concentrations; however, as Ca^2+^ levels increase, it becomes catalytically active due to Ca^2+^/CaM binding to the transiently disordered regulatory domain and Ca^2+^ binding to the EF hand domain on CnB, resulting in the loss of the autoinhibitory domain regulating the catalytic site [200,201,202]. CaN regulators and substrates share two possible short linear motifs (SLiMs) PxIxIT and LxVP, which binds to their binding pockets found on CnA and CnA/CnB, respectively [201,203]. The two main inhibitors of CaN-cyclosporin A (CsA) and FK506- has been shown to bind to the LxVP binding pocket as well [203]. Specific targeting of CaN with CsA and FK506 to suppress angiogenesis has been long explored; however, it is greatly limited by their side effects and toxicity [204]. Quercetin has been suggested as a potent antiangiogenic agent that targets CaN as a treatment for breast cancer [205]. In contrast to several publications, a recent study found that calcineurin inhibitors CsA and FK506 promote EC proliferation and a VEGFR-induced angiogenesis-like process. Moreover, endothelial-to-mesenchymal transition was identified as a result of inflammatory responses, which were previously attributed to CaN inhibition [206]. CsA also induces EC proliferation and migration through a CaN-independent pathway by releasing mitochondrial ROS [207]. CaN plays a direct role in cell cycle regulation, as inhibition of the CaN/nuclear factor of activated T-cell (NFAT) pathway by various CaN inhibitors causes cyclin D1 degradation by inhibiting the dephosphorylation of Thr286 of cyclin D1 [208]. However, the therapeutic use of CaN inhibitors in tumor angiogenesis is complex, as their potential pro-angiogenic effects must be balanced against their immunosuppressive properties and associated risks of de novo malignancies. Further research is needed to better understand the precise mechanisms of CaN in tumor angiogenesis and to optimize therapeutic strategies targeting this pathway.

CaN participates in various cellular processes such as T-cell activation, cell survival regulation, proliferation, migration, growth, angiogenesis, differentiation, transcription regulation, tumor progression, and metastasis [197]. Notably, the CaN/NFAT signaling pathway is significant, wherein CaN dephosphorylates members of the NFAT transcription factors upon Ca^2+^ signals, allowing the translocation of NFATs to the nucleus to regulate several target genes [209,210]. Additionally, noncanonical Wnt/Ca^2+^ signaling contributes to wound angiogenesis and repair. Secreted Frizzled-related protein-2 induces angiogenesis via the noncanonical Wnt/Ca^2+^ pathway, leading to NFATc3 dephosphorylation by CaN [211]. Studies using in vivo mouse Matrigel plug assays and chick chorioallantoic membrane assays demonstrated increased EC migration and tube formation, which could be reversed by Tacrolimus treatment in angiosarcoma xenograft models [211,212]. Research has also shown that Wnt-CaN-VEGFR-1 signaling is active in macrophages, where VEGFR-1 suppresses wound angiogenesis by binding VEGF-A [213]. CnA-binding protein (CnABP) was found to be coexpressed with paired box 2 (PAX2) in the developing kidney in vitro, a crucial gene involved in kidney development and also a key regulator of Wilm’s tumor progression. Yeast two-hybrid experiments identified CnABP as an interaction partner of CnAß. It was concluded that overexpression of CnABP is a negative regulator of calcineurin, which is quite surprising as both are positively implicated in cell proliferation and migration [214,215]. Regulator of calcineurin 1 (RCAN1) has been shown to have great importance in regulating calcineurin-related angiogenetic processes. VEGF-A upregulates RCAN1.4 expression in ECs by increasing intracellular Ca^2+^ concentrations via PLCγ phosphorylation, thus activating the calcineurin/NFAT pathway. RCAN1.4 upregulation by VEGF-A is also achieved by PKCδ activation in a calcineurin-independent mechanism [216,217]. Aurintricarboxylic acid was shown to promote VEGF-induced angiogenesis by disrupting plasma membrane calcium ATPase 4 (PMCA-4) calcineurin interaction and shows promising results of reperfusion in experimentally induced ischemia in in vivo animal models [218]. An important finding in the literature suggests that RCAN1.4 significantly influences endothelial cell migration and angiogenesis by controlling the internalization of VEGFR-2 [219]. In the pre-metastatic niche, a predetermined site for metastasis in the lung tissue, Ang-2 is transactivated due to increased VEGF signals, promoting angiogenesis and tumor seeding via the CaN/NFAT pathway [220]. RCAN1 expression was found to be significantly decreased in clear cell renal carcinoma, which is in correlation with poor prognosis and could explain its role in suppression of cancer progression [221]. CnAß is overexpressed in small cell lung cancer cells, significantly increasing cell proliferation and invasion, which is speculated to be responsible for bone metastasis [222]. Mammary cancer cell survival, migration, and invasion are also highly dependent on calcineurin-activated NFAT1 and NFAT2 signaling [223].

Recent findings suggest that CaN does not significantly impact angiogenetic sprouting and primary tumor growth in tumor angiogenesis. Instead, it modulates vascular regression and remodeling, restricting metastatic outgrowth and stabilizing vessels during the early stages of metastasis formation [224,225]. R-Spondin 3 (Rspo-3) has emerged as a key regulator of vascular remodeling through the noncanonical Wnt/Ca^2+^/CaN/NFAT signaling pathway [225]. These findings also imply that CaN inhibitors directly target ECs, in addition to immune cells, which may contribute to increased de novo malignancies following immunotherapy in solid organ transplant recipients [224].

Research on calcineurin inhibitors, particularly in the context of clinical trials, is an active area of investigation. While these clinical trials not directly focusing on cancer angiogenesis, indirectly addresses potentially influence angiogenic processes. Calcineurin inhibitors like tacrolimus and cyclosporine have been studied extensively for their immunosuppressive properties, particularly in the context of transplantation [226]. Mesenchymal stem cells combined with CD25 monoclonal antibody and calcineurin inhibitors or the usage of tacrolimus and methotrexate are tested to treat acute graft-versus-host disease [227,228].

The important pathways are summarized in Figure 5, which provides a schematic representation of the impact of PPPs on tumor angiogenesis.

## 4. Role of the Main Tyr Phosphatases in Tumor Angiogenesis

Protein tyrosine phosphatases (PTPs) play key role in regulation of cellular and physiological functions by governing signal transductions through dephosphorylation of tyrosine residues on signal proteins [229]. Adequate levels of tyrosine phosphorylation are ensured by a balanced function between kinases and phosphatases [229,230]. Since the numbers of tyrosine kinases and phosphatases are roughly equivalent, with 85 known and catalytically active tyrosine kinases suggesting the similar amount of PTPs [231]. This chapter discusses the small number of PTPs that have been identified as essential modulators of angiogenesis, offering an opportunity to discover novel therapeutic targets for obstructing angiogenesis, mainly in the context of developing tumor vasculature [230].

### 4.1. Vascular Endothelial Protein Tyrosine Phosphatase (VE-PTP)

VE-PTP is an endothelial-specific receptor-type tyrosine phosphatase that play an important regulatory role in embryonic and tumor angiogenesis [229,231] (Figure 6). In the development and integrity of the vascular system, Tie-2,a transmembrane tyrosine kinase receptor, and VE-cadherin, an endothelium-specific adhesion molecule, are defined as main downstream substrates for VE-PTP [229,231,232]. VE-PTP is highly selectively expressed in ECs, where it physically associates with cytoplasmic domain of the Tie-2 receptor to control its signaling [44,45,50]. VE-PTP plays an essential role in balancing and regulating Ang/Tie-2 signaling in the context of EC proliferation, vessel remodeling, permeability in embryonic and tumor angiogenesis, as well as in inflammation [53,231]. Binding of VE-PTP dephosphorylates and negatively regulates the angiogenic effects of Tie-2; therefore, genetic deletion or specific inhibition of VE-PTP results in Tie-2 activation [44,233]. Taking into consideration the presence or absence of cell–cell contacts, distinct Ang-1/Tie-2 signaling complexes can be assembled, thus activating different subcellular domains of Tie-2 and stimulating various patterns of signal transduction proteins [234]. In intercellular junctions, Ang-1 stimulates the association of VE-PTP with Tie-2, and this VE–PTP/Tie-2 complex reduces vascular EC permeability [234]. Other research has identified VE-PTP and Tie-2 in the presence of VEGFR-2, creating complexes that contain all three components. Their specific ligands, VEGF and Ang-1, recruit the participants of this complex to endothelial junctions. Stress-induced VEGF enhances the number of VE-PTP/VEGFR-2 complexes at endothelial junctions but reduces overall in the cell, possibly as a consequence of internalization and degradation. Both Tie-2 and VEGFR-2 require the dissociation of VE-PTP to gain their activity; hence VE-PTP has a “safeguard” function at endothelial junctions to maintain endothelial quiescence [235]. The second main downstream target of VE-PTP is the endothelium-specific adhesion molecule VE-cadherin, which connects ECs to each other, creating adhesion junctions and preserving integrity in resting cells [232]. Maintaining the endothelial barrier demands the association of VE-PTP with VE-cadherin in endothelial junctions. VE-PTP inhibits the tyrosine phosphorylation and downregulation of VE-cadherin [44,50]. Angiogenic stimuli induce high expression of VEGF, which triggers the dissociation of VE-PTP from VE-cadherin, thereby inducing the phosphorylation and internalization of VE-cadherin through activating the VEGFR-2-mediated signaling cascade [44,236]. During vascular quiescence, VE-PTP dephosphorylates and sustains the inactiveness of VEGFR-2, thus ensuring its stabilizing effect on endothelial junctions. During angiogenesis, the altered function and signaling of VE-cadherin are due to enhanced gene transcription and protein phosphorylation on tyrosine residues. In the presence of VEGF, VEGFR-2 connects to and phosphorylate VE-cadherin, inducing cell survival through PI3K/Akt axis, cell proliferation via ERK/MAPK pathway and cell motility via Cdc42 activation [232].

Over the years, several downstream targets of VE-PTP, such as the above mentioned VEGFR-2 or later discussed ephrin type B receptor 4 (EPHB4), have been discovered. RhoGEF And PH domain-containing protein 5 (FGD5), the GTPase exchange factor for Cdc42, was described as a direct VE-PTP substrate, contributing to the essential regulatory role of VE-PTP for endothelial junction integrity [231,237]. Upstream regulators of VE-PTP, nxhl (New XingHuo light gene) and nucleolin (NCL), were identified as regulators of angiogenesis. There is evidence of interactions between nxhl and NCL, as well as NCL and VE-PTP, creating a novel nxhl/NCL/VE-PTP axis for angiogenic control. The nxhl gene affects and controls NCL, which directly increases expression of VE-PTP. Angiogenesis is induced by nxhl via stimulating proliferation, migration, and tube formation. In addition, nxhl contributes to EMT and tumor invasion. These findings reveal that nxhl/NCL/VE-PTP signaling pathway can be a potential therapeutic target for cancer treatment [233]. Consequently, VE-PTP can offer a new therapeutic target in tumor angiogenesis [238].

AKB-9778 is a novel, potent, and selective inhibitor for pharmacological VE-PTP domain inhibition that has been developed and extensively investigated [237,239]. The effects of VE-PTP inhibition using AKB-9778 are mainly mediated via upregulated Tie-2 kinase activity, which strongly induces tyrosine phosphorylation of itself, Tie-2, and other downstream targets of VE-PTP, such as FGD5 and EPHB4 [237,239]. EPHB4 controls endothelial function and vessel specification, thereby playing an important role in vascular development. EPHB4 was identified as a component of a ternary complex with Tie-2 and VE-PTP [237]. Tie-2 activation is necessary for tumor angiogenesis by promoting vascularization and tumor growth. Therefore, inhibition of VE-PTP and its subsequent ligand-independent Tie-2 activation may raise many questions. Here, we must consider the dual role of Tie-2 in both the normal and tumor vasculature, as well as the regulation of Tie-2 by multiple ligands with opposing effects [238]. AKB-9778-induced Tie-2 activates downstream signaling through phosphorylation of Akt and eNOS, thereby inducing production of NO in ECs [238,239]. Increasing levels of intratumoral NO around the endothelial compartment normalize the structure and function of tumor vessel, optimizing perfusion and decreasing hypoxia [238]. Other research has shown that VE-PTP inhibition delays the early phase of primary tumor growth, metastatic progression, and prolongs survival [239]. VE-PTP inhibition-induced structural and functional normalization of vessels corresponds with reduced vascular permeability and micrometastasis of tumor cells, which would be prevented by tight junctions of ECs. Reduced hypoxia due to Tie-2 activation improves the efficacy of radiotherapy and chemotherapy since tumor requires oxygen to respond to radiation and cytotoxic agents to achieve maximal tumoricidal effects [238,239].

While there are no current clinical trials specifically targeting VE-PTP in cancer therapy, the preclinical evidence provides a strong rationale for future clinical development. Inhibition of VE-PTP enhances Tie2 signaling, promoting vascular stability and reducing aberrant angiogenesis [53,240,241]. This provides a therapeutic angle where modulating VE-PTP activity can suppress tumor angiogenesis.

### 4.2. Human Receptor-Type PTP Kappa (PTPRK)

Human receptor-type PTP kappa (PTPRK or R-PTP-kappa) is expressed by various normal cell types, including vascular ECs; however, several cancer cell lines show aberrant expression of PTPRK [229,242]. So far, only little information is available about the precise biological functions of PTPRK, and its role in tumor-associated angiogenesis remains unclear [229,242]. PTPRK serves as an essential coordinator of cell signal transduction from several receptors depending on phosphorylation of tyrosine residues, which may explain its potential role in angiogenesis [229]. Knockdown of PTPRK in vascular ECs has provided insight into its dual roles in regulating angiogenesis. PTPRK enhances cell-matrix adhesion, presumably through FAK-paxillin signaling pathway in ECs, coinciding with similar observations in prostate cancer cells [229]. Although PTPRK supports normal tubule formation, it seems to inhibit tubule formation by suppressing pathways induced by VEGF, FGF, and cancer cells. Therefore, a positive role of PTPRK is supposed in regulating cancer-cell induced angiogenesis. PTPRK inhibits migration, especially FGF-promoted migration of the ECs, and this inhibitory effect was previously observed in breast cancer cells [229]. Such opposing effects of PTPRK in different cancers suggest a cancer-specific role. However, these contrasting effects appear in vascular ECs, indicating more complex functions of PTPRK by interacting with different pro-angiogenic factors [229]. Several studies suggest the tumor suppressor role of PTPRK, further supported by its association with TGF-β and EGFR signaling pathways [242,243,244]. PTPRK is upregulated by TGF-β via stimulating the binding of Smad3 and Smad4 transcription factors to the PTPRK gene promoter. PTPRK specifically dephosphorylates EGFR, hence inhibiting cell proliferation and migration caused by EGF-induced EGFR-dependent signaling [242,243,244]. Besides EGFR, another identified substrate for PTPRK is β-catenin, which has undisputed role in angiogenesis [245,246,247]. PTPRK can be associated with neuropilin 1 (NRP1), a non-tyrosine kinase transmembrane glycoprotein. NRP1 overexpression in tumors and contribution to tumor angiogenesis were evidenced excepting for the poorly clarified adherent molecular mechanism [248].

### 4.3. Protein Tyrosine Phosphatase Receptor Type J (PTPRJ)/Density-Enhanced Phosphatase-1 (DEP-1)

PTPRJ, also known as DEP-1 (junction-associated density-enhanced phosphatase-1), CD148, PTP-eta, or HPTPη, belongs to the receptor-type PTPs [249]. The name “density-enhanced phosphatase” originates from its role in density-mediated growth inhibition, as increasing cell density correlates with elevated PTPRJ expression [249]. Among various cell types, vascular ECs notably express PTPRJ, which was initially identified as a tumor suppressor gene due to its involvement in negatively regulating several protein tyrosine kinases, including PDGFR, EGFR, VEGFR-2, and HGFR, thereby controlling angiogenesis, cell proliferation, and migration. Markedly decreased protein levels of PTPRJ have been observed in cancer cell lines (e.g., colon, breast, pancreas, thyroid, and lung), suggesting the tumor-suppressive role of PTPRJ [249]. However, significantly higher upregulation of PTPRJ has been identified in glioblastoma multiforme, indicating a potentially cancer-specific function of PTPRJ [249,250]. PTPRJ plays a negative role in regulating EC growth by inhibiting VEGF-dependent ERK1/2 activation [251,252]. Upon VEGF stimulation, PTPRJ dephosphorylates and attenuates VEGFR-2 activity, resulting in the subsequent downregulation of the proliferative PLCγ-ERK1/2 signaling cascade [251,252]. Furthermore, VE-cadherin inhibits VEGFR-2 proliferative signaling by binding and retaining VEGFR-2 at the cell surface, thereby preventing its endocytosis and favoring its inactivation by PTPRJ [5]. In gastric cancer, PTPRJ inhibits malignant transformation by dephosphorylating EGFR and blocking the mitogen-activated protein kinase (MEK)/ERK and PI3K/Akt pathways. In cervical cancer cell lines, PTPRJ inactivates the Janus kinase 1 (JAK1)/STAT3 pathway, impeding proliferation and tumor formation [250]. Soluble TSP-1 binds to the extracellular domain of PTPRJ, enhancing its catalytic activity and inhibiting cell growth. Given that TSP-1 is produced by various cell types in pathological conditions, including endothelial or cancer cells, TSP-1 appears to act as a ligand for PTPRJ in such conditions [253].

During retinal vascular development, PTPRJ-dependent upregulation of the Src/Akt pathway activates the DLL4-Notch-signaling pathway, resulting in the restriction of cell proliferation through Akt-mediated inhibition of rapidly accelerated fibrosarcoma (Raf), an upstream regulator of ERK1/2, and Notch-mediated inhibition of ERK1/2 [251]. Conversely, in ECs, PTPRJ plays a positive role in regulating permeability and angiogenesis via VEGF-dependent Src activation [254]. In the context of retinal vascular development, PTPRJ contributes to the initiation of VEGF-induced capillary formation and vascular permeability through the indirect induction of the DLL4/Notch signaling pathway. PTPRJ induces Src/Akt pathway activation in response to VEGF stimulation, leading to the phosphorylation and nuclear translocation of β-catenin. The resulting β-catenin transcriptional complex induces DLL4 expression, which interacts with the Notch receptor on neighboring cells, inducing capillary sprouting [251]. Several studies have confirmed PTPRJ-mediated Src activation. PTPRJ dephosphorylates the inhibitory Tyr residue of Src, leading to Src auto-phosphorylation and full activation, which can initiate tumor-associated angiogenesis [250,251,252,254]. A short variant of PTPRJ, named sPTPRJ, is generated by alternative splicing, resulting in a soluble protein secreted into the supernatant by both endothelial and tumor cells. Unlike PTPRJ, sPTPRJ expression is not subject to density-dependent regulation. The proangiogenic activity of sPTPRJ is manifested in the downregulation of endothelial adhesion molecules, increased tube formation, and EC migration. Moreover, in the most angiogenic tumors, such as human high-grade glioma, increased sPTPRJ mRNA levels have been observed compared to controls [252].

While not exclusively focused on angiogenesis, a clinical trial includes the investigation of PTPRJ genetic variations in cancer patients, which may provide insights into its role in tumor angiogenesis and potential therapeutic targeting.

### 4.4. Protein Tyrosine Phosphatase Receptor Type O (PTPRO)

PTPRO has tumor suppressive role in several cancers [255,256]. The downregulation of PTPRO in cancers results from its epigenetic silencing, caused by hypermethylation of the promoter region of the PTPRO encoding gene [255,256]. Upregulating PTPRO expression in tumors, such as esophageal squamous cell carcinoma, breast, and colorectal cancers, has tumor-suppressive effects by preventing tumor growth and metastases. These positive effects may be attributed to induced apoptosis and suppressed tumor-associated angiogenesis [255,256,257]. Research in colorectal cancer has revealed that PTPRO can associate with erb-b2 receptor tyrosine kinase 2 (ERBB2), thereby inhibiting signaling pathways and obstructing tumor growth [255]. Furthermore, PTPRO could inhibit colorectal cancer development and metastasis by regulating the Akt/mTOR/sterol regulatory element binding protein 1 (SREBP1)/acetyl-CoA carboxylase 1 (ACC1) and MAPK/peroxisome proliferator-activated receptor alpha (PPARα)/peroxisomal straight-chain acyl-CoA oxidase 1 (ACOX1) signaling pathways, thereby reprogramming lipid metabolism [255].

### 4.5. Protein Tyrosine Phosphatase Receptor Type Z (PTPRZ)

PTPRZ, also known as RPTPzeta, RPTPz, PTPζ, or RPTPβ, belongs to the subfamily of receptor-type PTPs [258,259]. PTPRZ expression is widespread during embryonic development but becomes restricted during adulthood [260]. Although the human PTPRZ1 gene encodes only the PTPRZ enzyme, its transcription produces 16 different mRNAs, including 14 alternatively spliced variants, 2 unspliced forms, and 3 known functionally identified splicing variants (PTPRZ-A, -B, and -S) [258,259,260]. Beyond alternative splicing, different PTPRZ forms can be derived from proteolytic processing [259,260]. Plasmin, TNF-α-converting enzyme, and MMP-9 cleave the extracellular domain of PTPRZ at multiple sites, releasing detectable extracellular PTPRZ fragments [259,260]. Therefore, PTPRZ proteins are found in the cell membrane, cytoplasm, and nucleus, but we limit this chapter to transmembrane PTPRZ, which interacts with several cell adhesion molecules, such as integrins and nucleolin, likely responsible for PTPRZ translocation to the nucleus [259]. PTPRZ has been extensively researched in the central nervous system, where it is primarily expressed, but its expression has also been detected in several other tissues and in ECs. Moreover, overexpressed PTPRZ has been observed in several types of cancer [259,260,261]. PTPRZ expression is selectively stimulated by HIF-2α but not by HIF-1α, suggesting tissue- or cell-specific (such as ECs) expression preference according to HIF-2α predominance [259]. PTPRZ has a significant stimulatory role in cell migration not only in neurons but also in endothelial and various tumor cells [259]. The first identified soluble ligand of PTPRZ was pleiotrophin (PTN), a secreted heparin-binding growth factor highly expressed in the embryonic nervous system and other embryonic organs, and overexpressed in several cancer cell lines [259]. PTN, also known as p18, heparin-binding growth factor-8/growth-associated molecule/neurotrophic factor, heparin affin regulatory peptide, or osteoblast-specific factor 1, has been identified as an angiogenic factor with an important role in cell growth, migration, and angiogenesis. PTN can influence endothelial and cancer cell functions through numerous cell surface receptors, including PTPRZ [259]. In ECs, PTN binding to PTPRZ results in c-Src dephosphorylation and activation, subsequently stimulating downstream targets such as integrin ανβ3, FAK, xanthine oxidase, PI3K, and ERK1/2 [259]. The integrin α_ν_β_3_ receptor is upregulated during angiogenesis in ECs [262]. The PTN/PTPRZ/Src axis-mediated α_ν_β_3_ receptor phosphorylation is required for the cell surface localization of NCL, which plays a critical role in tumorigenesis and angiogenesis [261]. PTN-induced PTPRZ activation of β-catenin induces cell migration, proliferation, and VEGF expression, leading to angiogenesis [259,260]. Another ligand of PTPRZ is FGF-2, and furthermore, VEGF-A can directly interact with PTPRZ, competing with PTN for PTPRZ binding [259,260,261]. VEGF induces similar downstream signaling pathways to PTN through PTPRZ, including Src, FAK, PI3K, MAPK, PTEN, MEK, ERK1/2, and integrin α_v_β_3_ activity [260,263]. VEGF interaction with PTPRZ induces c-Src-mediated α_v_β_3_ phosphorylation, required for both cell surface nucleolin localization and enhanced α_ν_β_3_ -VEGFR-2 interaction, resulting in VEGF-induced EC migration and angiogenesis [261]. The downstream VEGF effects via VEGF-PTPRZ interaction are not affected by anti-VEGF drugs, which may explain the resistance developed by certain tumor types (such as glioblastoma) to anti-angiogenic cancer therapies [261].

### 4.6. Phosphatase of Regenerating Liver-3 (PRL-3)

PRL-3 is an intracellular phosphatase expressed in embryonic blood vessels, pre-erythrocytes, and fetal hearts but it is absent in their mature counterparts. In some human colon samples, PRL-3 was detected in a subset of ECs [264]. Several studies have reported highly expressed PRL-3 in the main tumor mass, such as colorectal, gliomas, breast, ovarian, or gastric cancer; however, PRL-3 overexpression was significantly more pronounced in the metastasis of cancer, suggesting the involvement of PRL-3 in the metastasizing process [265,266]. This indicates that the PRL-3 protein plays an essential role in early angiogenic events, but at a later time, higher expression of PRL-3 correlates with tumor angiogenesis and progression [265]. Some PRL-3 expressing metastatic tumors expand inside blood vessels [264]. PRL-3 presumably initiates tumor angiogenesis and provokes cancer metastasis through the recruitment of ECs into the tumor areas to deliver nutrition and promote tumor growth [264,265]. An expected mechanism is the responsiveness of ECs to factors secreted by PRL-3 expressing cells [264]. One of the main pro-angiogenic factors, VEGF, is highly expressed in PRL-3 overexpressing cancer cells [265]. PRL-3 induces VEGF-A expression by promoting ERK phosphorylation and increasing the expression and activation of RhoGTPase, Rho-A, and Rho-C. Phosphorylated ERK translocates into the nucleus, promoting VEGF gene transcription and managing vascular formation. PRL-3 also induces VEGF-C overexpression to promote lymphatic vessel formation and accelerate the lymphatic metastasis of lung cancer [266,267,268]. PDGF, a key component of vascular remodeling and angiogenesis, induces Src-mediated phosphorylation of PRL-3, a modification necessary for PRL-3 mediated cell migration and invasion [268]. Furthermore, PRL-3 inactivates C-terminal Src kinase (Csk), a negative regulator of Src [268]. Hence, PRL-3 indirectly activates Src kinase, a proto-oncogene non-receptor tyrosine protein kinase, which can activate a series of signal transduction pathways, including PDGFR-β, ERK1/2, STAT3, or FAK [268,269,270]. Consequently, FAK activates the p38 and PI3K/Akt pathway, accomplishing the oncogenic effect of PRL-3 [268]. PRL-3 may play an important role in EMT by inhibiting the PI3K inhibitor, PTEN, which activates Akt, then inactivates GSK-3 following a subsequent overexpression of mesenchymal markers—vimentin, fibronectin, and Snail—and downregulation of E-cadherin, β-catenin, and integrin [265]. Research involving colorectal cancer cells has validated that PRL-3 potentially triggers angiogenesis by elevating the secretion of TNFα and activating the NFκB pathway [268,271]. This contributes to enhanced production of proinflammatory cytokines interleukin (IL)-6 and IL-8, promoting metastasis [268,271]. Furthermore, PRL-3 reduces IL-4 secretion, forcing the vasculogenesis at another point [264]. Another study on hepatocellular carcinoma revealed a strong correlation between PRL-3 and high levels of MMP-2 and MMP-9 expression, directly linked to the tumor’s angiogenic capacity in experimental animal models [269]. The degradation of extracellular matrix protein facilitates cancer cells to invade the surrounding tissues, thereby creating metastases. Presumably, PRL-3 activated ERK1/2 and PI3K/Akt pathways mediate the induction of MMP-2 and MMP-9 expression [269].

PRL-3 participates in numerous cancers and intricate signal transduction pathways, indicating that targeting this phosphatase could be a potential strategy in anti-tumor therapy [267]. TGFβ serves as one of the regulators of PRL-3 expression at the transcriptional level, operating through a pathway involving Smad3. Another regulator, polyC-RNA-binding protein 1 (PCBP1), disrupts PRL-3 mRNA transcription. Thienopyridone (7-amino-2-phenyl-5H-thieno [3,2-c] pyridine-4-one), a compound used in antiplatelet drugs, can inhibit PRL-3 as well as its downstream target p130Cas. Natural compounds like curcumin from turmeric and the methanolic extract of Rubia akane roots selectively block PRL-3. Additionally, antibody-based therapy shows promise as an anticancer treatment strategy targeting PRL-3 [265]. A phase I study evaluates the safety and efficacy of PRL-3-Zumab, a monoclonal antibody targeting PRL-3, in patients with advanced solid tumors [272]. PRL-3′s role in angiogenesis and metastasis makes it a potential target for inhibiting tumor vascularization.

### 4.7. Src Homology 2 (SH2) Domain-Containing Protein Tyrosine Phosphatase-1 (SHP-1)

SHP-1, also known as PTPN6, is a non-receptor protein tyrosine phosphatase primarily localized in the cytoplasm and expressed predominantly in hematopoietic lineages and ECs [229,273,274]. Recognized as a tumor suppressor, SHP-1 negatively regulates growth factor receptors by dephosphorylation [229,274,275]. Receptor tyrosine kinases like EGFR and VEGFR have been found to harbor binding sites for SHP-1, facilitating their negative regulation [276,277]. Upon recruitment and activation of SHP-1, receptor dephosphorylation occurs, alongside dephosphorylation of the SH2-domain binding sites, consequently disrupting the complex [277]. SHP-1 plays a role in repressing angiogenesis by obstructing VEGF signaling in microvascular ECs [2]. Cluster of differentiation (CD) 36, a receptor for TSP-1, is crucial in the regulation of angiogenesis and pathological neovascularization [276]. Interaction between CD36 and TSP-1 recruits SHP-1 to the VEGFR-2 signaling complex, diminishing VEGF signaling, a well-known proangiogenic factor influencing EC function [276]. In various cancer types, SHP-1 has been identified as a negative regulator of STAT3, which directly controls VEGF transcription [275]. Tissue inhibitors of metalloproteinases (TIMPs) regulate MMP activity essential for cancer progression, metastasis, and angiogenesis, independent of their MMP inhibitory activity [278]. Among TIMPs (TIMP 1–3), TIMP-2 interacts with the integrin α_3_β_1_ receptor on the surface of human microvascular ECs. This interaction initiates an SHP-1-induced signal cascade that maintains endothelial quiescence. SHP-1 inactivates angiogenic receptors like FGFR-1 and VEGFR-2, suppressing EC proliferation and angiogenesis [278]. Additionally, SHP-1 activates the small GTPase Rap1 signaling, leading to repression of EC migration, and induces the cyclin-dependent kinase inhibitor p27^Kip1^, blocking the cell cycle [278]. Through multiple pathways, SHP-1 exerts its anti-angiogenic effect, positioning it as a promising therapeutic target in angiogenesis [278].

### 4.8. SH2 Domain-Containing Protein Tyrosine Phosphatase-2 (SHP-2)

SHP-2, also known as PTPN11, SH-PTP2, SH-PTP3, PTP2C, PTP1D, or Syp, is a ubiquitously expressed non-receptor tyrosine-specific enzyme [279,280,281]. While SHP-2 shares a homologous structure and sequence with the tumor suppressor SHP-1, it is the first reported oncogenic PTP [279]. SHP-2 can directly bind to the intracellular domains of auto-phosphorylated receptor tyrosine kinases such as EGFR, PDGFR, and VEGFR, or indirectly via its SH2 domains using tyrosine-phosphorylated adaptor proteins [279,280]. During hematopoietic cell development, SHP-2 plays a balancing role between apoptosis, proliferation, and differentiation. Endothelial SHP-2 is particularly crucial for initiating angiogenesis and new blood vessel formation by preventing EC apoptosis [281]. Upregulation of SHP-2 can contribute to tumorigenesis by increasing tumor angiogenesis [282]. Additionally, SHP-2 promotes cell migration and invasiveness, and its overexpression is associated with induced cell proliferation, increased clone formation, and decreased chemotherapeutic sensitivity [283]. Given its crucial role in mediating signal transduction of growth factor receptors such as EGFR, FGFR, and VEGFR, and in enhancing other tumorigenic pathways like Ras, Src, Akt, and ERK, overexpression or gain-of-function mutations of SHP-2 are associated with multiple human cancers [279,280,282]. SHP-2 also plays an important role in angiogenesis and endothelial permeability by controlling endocytosis-mediated VE-cadherin cycling [281]. ANXA2, a scaffolding protein, contributes to SHP-2-mediated dephosphorylation of VE-cadherin in angiogenesis [284]. Growth factor receptor bound protein 2 (GRB2) associated binding protein 1 (GAB1), another well-known scaffolding protein for SHP-2, promotes postnatal angiogenesis. GAB1 associates with SHP-2 during EGF, FGF, and VEGF stimulation, mediating signal transduction [279,281,284]. However, SHP-2 can also interact directly with VEGFR-2 and induce angiogenic VEGFR-2 signaling [284,285]. SHP-2 dephosphorylates RasGAP, a negative regulator of Ras, and the negative Src regulator Csk, thereby activating Ras and Src signaling pathways [279]. Src activation induces dynamin II–dependent VEGFR-2 receptor internalization, thereby promoting VEGFR-2 internalization and its angiogenic signaling [285]. Processes essential for angiogenesis initiation, EC survival, and proliferation are induced by SHP-2 primarily through the activation of MAPK (mitogen-activated protein kinase) over and above PI3K/Akt and ERK1/2 via Src and Ras activation [280,281]. SHP-2 can directly associate with the regulatory p85 subunit of PI3K and with GAB1 upon growth factor stimulation, suggesting that PI3K, SHP-2, and GAB1 create a signaling complex to activate PI3K and Akt [281]. Upregulated SHP-2 enhances cell migration and motility by inducing ERK, PI3K/Akt, and small GTPase pathways upon integrin or fibronectin stimulation [282]. Interaction between gap junction protein connexin 43 (Cx43) and SHP-2 has been reported to be responsible for EC migration and changes in the actin cytoskeleton [286]. SHP-2 regulation has been reported in MAPK cellular signal transduction through ERK, JNK/stress-activated protein kinase (SAPK), p38/ERK, and big mitogen-activated protein kinase 1 (BMK1)/ERK5 in leukemic and breast cancer cells. SHP-2-induced mammary tumorigenesis occurs by promoting ERK and PI3K signaling [283]. However, contradictory results have been reported regarding SHP-2′s regulation of ERK and p-38-MAPK pathways in ECs under hypoxic conditions. Instead, a proangiogenic apoptosis signal-regulating kinase 1 (ASK1)/c-Jun/SRY-Box transcription factor 7 (SOX7) signaling pathway was induced by SHP-2, responsible for tumor angiogenesis and vessel abnormalization. SHP-2 inefficiency results in tumor vascular normalization and impairs EC proliferation, migration, as well as tubulogenesis [284]. Under hypoxia, proangiogenic SOX7 is induced in ECs, resulting in hypoxia-induced angiogenesis, particularly in tumors. SHP-2 dephosphorylates and stabilizes ASK1, thereby stimulating downstream JNK-c-Jun signaling, which regulates SOX7 transcription, contributing to pathological angiogenesis [284]. SHP-2 is essential for hypoxic angiogenesis by preventing HIF-1α degradation via Src activation. HIF-1α induces the expression of several angiogenic genes under hypoxia [287]. SHP-2 has a key role in regulating tumor angiogenesis and vessel abnormalization by inducing multiple proangiogenic signaling pathways [284]. The high sequence similarity between SHP-2 and SHP-1 can pose challenges with the selectivity of SHP-2 inhibitors [279]. However, recently, SHP-2 allosteric inhibitors (e.g., SHP099) have become relatively easy to access [284]. SHP099 stabilizes SHP-2 in an inactive conformation, suppressing proliferation by inhibiting the Ras-ERK pathway [280]. SHP-2 is a promising target for anti-angiogenic cancer therapy [284] (Figure 7).

Several SHP-2 inhibitors are being tested in clinical trials in advanced solid tumors. TNO155 and RMC-4630, two SHP-2 inhibitors, are tested in phase I trials. By inhibiting SHP-2, these studies may provide insights into angiogenic signaling pathways, which are critical for tumor growth. Activation is indicated by pointed arrows, whereas flat arrows represent inhibition.

### 4.9. Protein Tyrosine Phosphatase Non-Receptor Type 12 (PTP-PEST)

PTP-PEST, encoded by the PTPN12 gene, is a ubiquitously expressed phosphatase containing PEST motifs (peptide sequence rich in proline (P), glutamic acid (E), serine (S), and threonine (T))that play an essential role in protein-protein interactions, allowing the enzyme to regulate its substrates or serve as a signal for protein degradation [288,289]. It is primarily expressed in hematopoietic cells [289]. PTP-PEST plays a crucial role in cell-matrix interactions, particularly during embryogenesis [290]. Interacting with key components of the FAK pathway, such as Src homology and collagen (Shc), paxillin, and GRB2, PTP-PEST contributes to EC survival and migration [229,290]. During cell spreading, PTP-PEST near the membrane regulates Rac1 small GTPase, inducing membrane ruffling and the formation of lamellipodia and focal adhesion complexes [290]. PTP-PEST may participate in embryonic angiogenesis through its interaction with GRB2, which can link PTP-PEST to the Ang/Tie-2 signaling pathway [290]. PTP-PEST is essential for developmental angiogenesis by inducing EC migration and vascular development [288,289]. Additionally, it can contribute to tumor angiogenesis by inducing AMPK dephosphorylation and activation, thereby promoting endothelial autophagy in response to hypoxia. Under hypoxic conditions, upregulated PTP-PEST induces EC migration, capillary tube formation, and autophagy, which are prerequisites for angiogenesis [288]. Hypoxia can induce PTP-PEST expression and activity through putative HIF-1α-induced transcription of PTP-PEST, as suggested by the presence of HIF-1α binding sites in the PTP-PEST promoter region. PTP-PEST also interacts with other proteins (e.g., ubiquitin thioesterase, protein kinase C-ξ, myotubularin related protein 6, and sarcolemmal membrane-associated protein), contributing to autophagy or its regulation [288].

### 4.10. Protein-Tyrosine Phosphatase 1B (PTP1B)

PTP1B, also known as PTPN1, is a widely expressed non-receptor type of tyrosine-specific PTP [273,279,291]. PTP1B plays a role in growth factor signaling and can both inhibit and promote tumorigenesis [273]. Its tumor suppressor function is evident in its ability to inhibit migration and invasion in fibroblasts and ECs by dephosphorylating and inactivating the Rac activator, p130cas. The Rac signaling pathway promotes actin polymerization, lamellipodia formation, and stimulates MMPs (MMP-2 and -7), which induce metastasis [273]. In ovarian cancer, PTP1B negatively regulates IGFR (insulin-like growth factor receptor) signaling, which is associated with survival and metastasis [273]. Macrophages play a tumor-supporting role after being recruited to neoplastic tissues. Activated macrophages produce several pro-angiogenic factors (such as VEGF) and proteases for extracellular matrix degradation. PTP1B dephosphorylates and negatively regulates CSF1R (colony stimulating factor 1 receptor), inhibiting macrophage activation and associated angiogenesis [292]. CSF1R regulation involves controlling the recruitment of the GRB2/GAB2/SHP-2 complex and modulating ERK signaling in macrophages [292]. Furthermore, PTP1B downregulates receptor tyrosine kinases, including EGFR, IGFR, PDGFR, FGFR, HGFR, and macrophage CSF1R, suggesting its involvement in various cellular regulations, including angiogenesis [292,293]. PTP1B stabilizes VE-cadherin-mediated cell-cell adhesions by reducing tyrosine phosphorylation of VE-cadherin through direct dephosphorylation and inactivation of VEGFR-2 [294]. PTP1B-driven dephosphorylation of endothelial VEGFR-2 interferes with VEGFR-2 internalization and complete activation of the ERK signaling cascade, highlighting the important role of PTP1B in angiogenesis [295]. However, PTP1B functions as an oncogene in many cancers, such as ovarian, gastric, prostate, pancreatic, and breast cancers [273,279]. PTP1B overexpression induces aggressive proliferation and migration by modulating the pyruvate kinase M2 pathway [273]. PTP1B can act as a tumor promoter in glioblastoma-multiforme, breast, colorectal, non-small cell lung, and ovarian cancer by activating Src and downregulating PTEN, thereby inducing downstream PI3K/Akt and Ras/Raf/MEK/ERK signaling, resulting in increased migration, invasion, and proliferation [273,279]. PTP1B also promotes tumorigenesis in breast cancer by influencing the JAK/STAT pathway [273]. In many aspects, PTP1B could be considered a promising cancer-specific molecular target, but its controversial nature may pose future complications. Although primarily focused on metabolic disorders, a trial investigates ERB-041, which has implications for PTP1B inhibition. Given PTP1B’s role in angiogenic signaling pathways, insights from this trial may be extrapolated to cancer therapy.

The key pathways are illustrated in Figure 8, which offers a schematic overview of how PTPs influence tumor angiogenesis.

## 5. Discussion

Tumor angiogenesis is a multifaceted process that is critical for tumor growth, metastasis, therapeutic resistance, immune evasion, and prognosis. While much attention has been given to the role of protein kinases in angiogenesis, the significance of protein phosphatases in this context is increasingly recognized. By dephosphorylating angiogenic receptors and downstream signaling proteins, protein phosphatases exert precise control over angiogenic signaling cascades, influencing processes such as endothelial cell proliferation, migration, and tube formation. Dysregulation of protein phosphatases in endothelial cells can lead to aberrant angiogenesis, contributing to tumor progression and metastasis. Targeting protein phosphatases has emerged as a promising approach for anti-angiogenic therapy in cancer. By selectively inhibiting the activity of specific protein phosphatases involved in angiogenesis, it may be possible to disrupt tumor vascularization and inhibit tumor growth and metastasis. Understanding the intricate crosstalk between these signaling pathways is crucial for developing targeted therapies to inhibit tumor angiogenesis selectively.

## Figures and Tables

**Figure 1 ijms-25-06868-f001:**
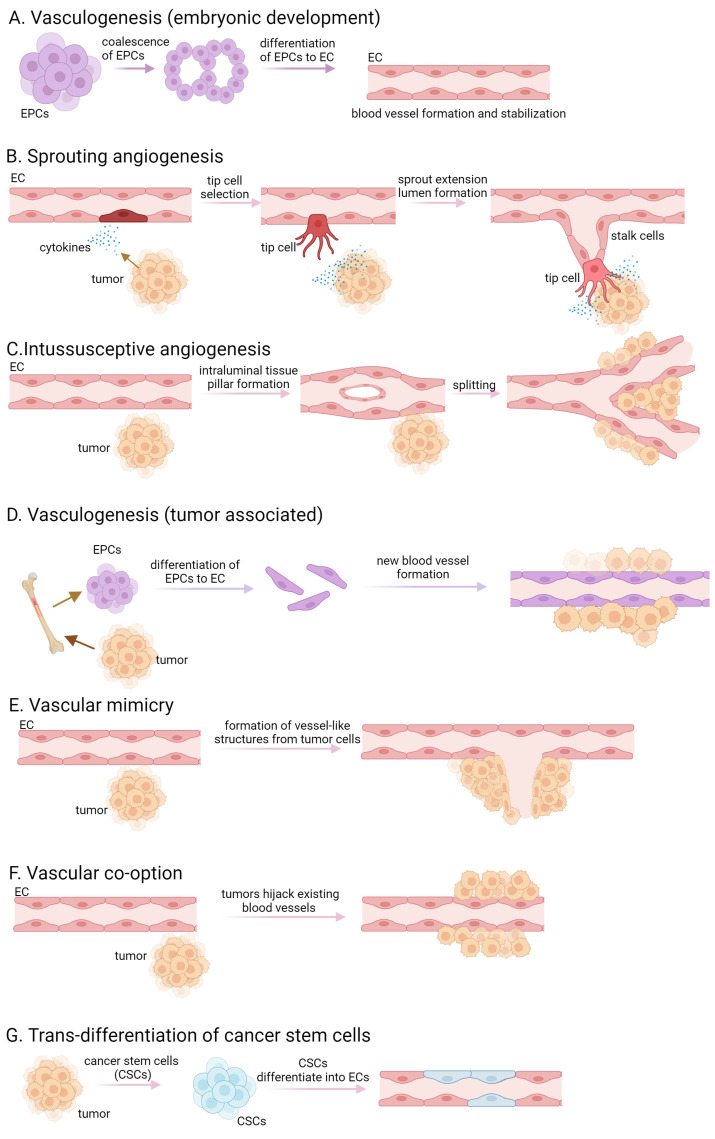
Different forms of vasculogenesis and tumor angiogenesis. (**A**) embryonic vasculogenesis; (**B**) sprouting angiogenesis; (**C**) intussusceptive angiogenesis; (**D**) tumor associated vasculogenesis; (**E**) vascular mimicry; (**F**)vascular co-option; (**G**) transdifferentiation of cancer stem cells.

**Figure 2 ijms-25-06868-f002:**
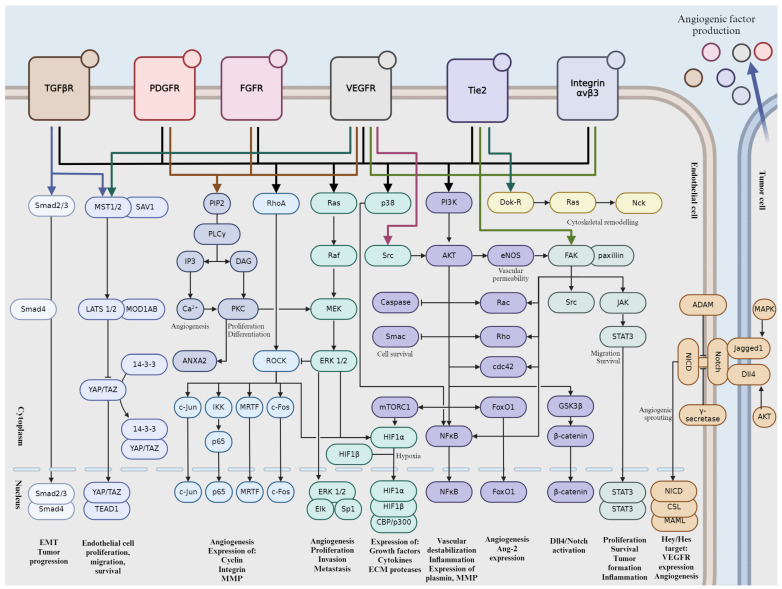
Schematic representation of signaling pathways during tumor angiogenesis. Activation is indicated by pointed arrows, whereas flat arrows represent inhibition.

**Figure 3 ijms-25-06868-f003:**
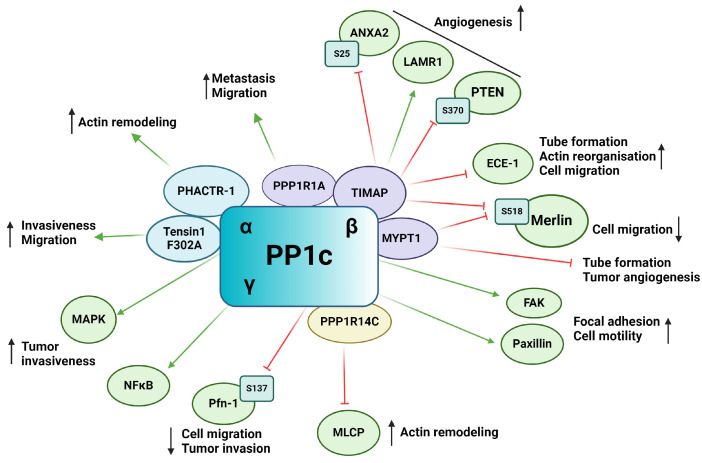
Schematic representation of the impact of PP1 on tumor angiogenesis via its substrates and interacting partners. Activation is indicated by green arrows, whereas inhibition is represented by red flat arrows.

**Figure 4 ijms-25-06868-f004:**
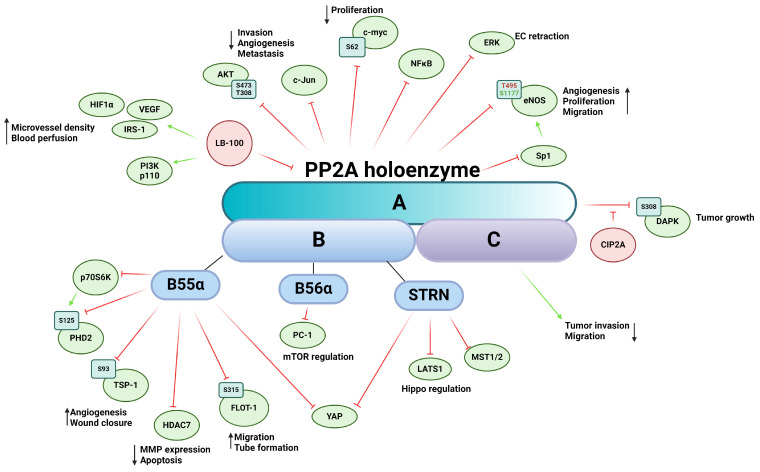
Schematic representation of PP2A’s impact on tumor angiogenesis via its substrates and interacting partners. The figure depicts the PP2A holoenzyme, highlighting its three subunits: the scaffold subunit A, the regulatory subunit B, and the catalytic subunit C. Activation is indicated by green arrows, whereas inhibition is represented by red flat arrows.

**Figure 5 ijms-25-06868-f005:**
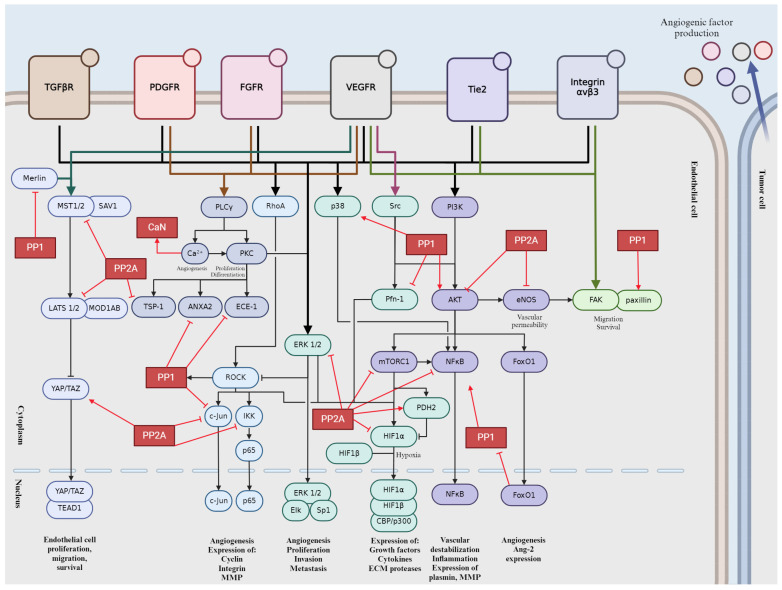
Schematic representation of PPPs in tumor angiogenesis signaling pathways. Activation is indicated by pointed arrows, whereas flat arrows represent inhibition.

**Figure 6 ijms-25-06868-f006:**
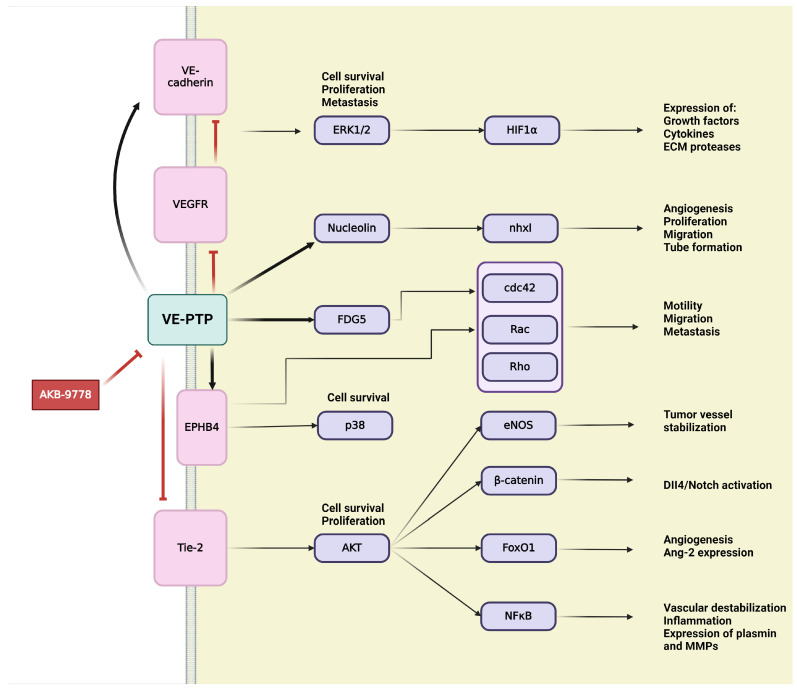
Schematic representation of VE-PTP signaling in tumor angiogenesis. Activation is indicated by pointed arrows, whereas red flat arrows represent inhibition.

**Figure 7 ijms-25-06868-f007:**
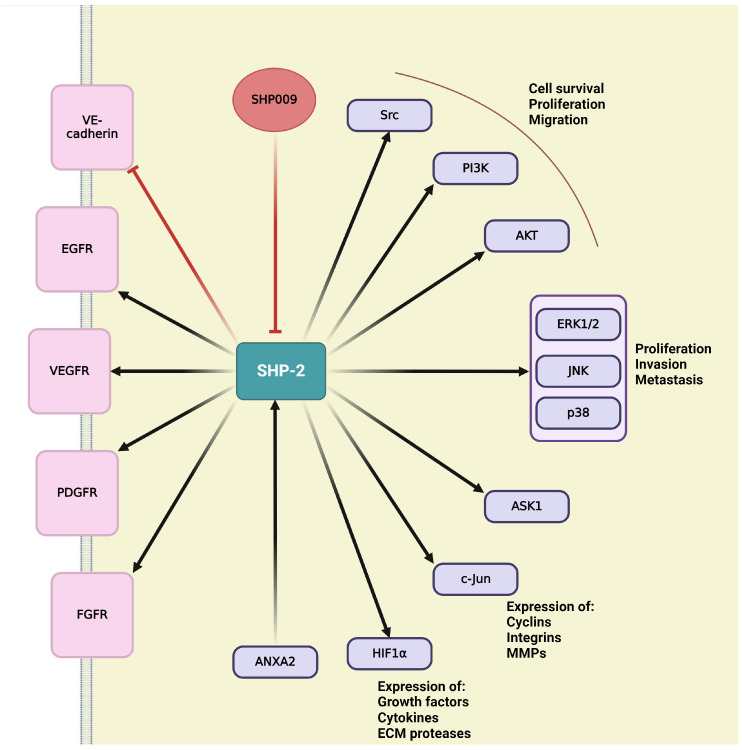
Schematic representation of SHP-2 signaling in tumor angiogenesis.

**Figure 8 ijms-25-06868-f008:**
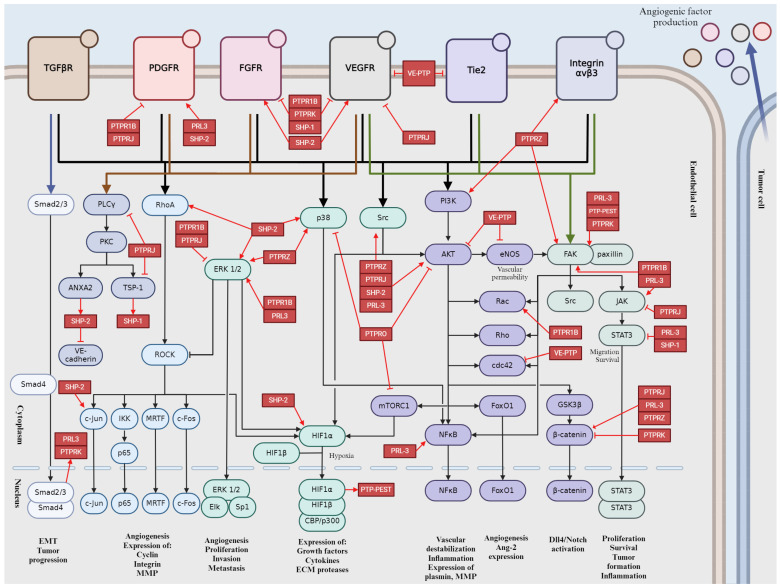
Schematic representation of PTPs in tumor angiogenesis signaling pathways. Activation is indicated by pointed arrows, whereas flat arrows represent inhibition.

## Data Availability

Data available in a publicly accessible repository.

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
