# Peer review of "Role of Protein Phosphatases in Tumor Angiogenesis: Assessing PP1, PP2A, PP2B and PTPs Activity"

_ijms, 2024, doi:10.3390/ijms25136868_

Round 1

Reviewer 1 Report

Comments and Suggestions for Authors

The manuscript entitled Role of Protein Phosphatases in Tumor Angiogenesis: As-2 sessing PP1, PP2A, PP2B, and PTPs Activity gives a detailed overview of the signaling pathways of angiogenesis as potential targets for anticancer therapeutical approaches. The review article is very well written, with very few number of mistyping or grammatical failures. The concept is clear and consistent and the topic is actual and is in the scope of the most modern pharmaceutical research fields in cancer science. The significance of phosphatases is emphasized in the manuscript. The figures are well-presented, detailed, and consistent with the written text.  

The article is based on 290 international articles published in relevant scientific journals. There are a low number of self-citations, which are within the acceptable rate.

I suggest extending the Discussion part with examples from the literature that support the potentiality of phosphatases as therapeutical targets in cancer therapy. Please, add original research articles that deal with the phosphatases in angiogenesis as molecular targets for new antitumor agents. Are there any preclinical or clinical data available?

Comments on the Quality of English Language

The quality of English in the manuscript is excellent.

Author Response

Thank you for your thorough and positive review of our manuscript. We appreciate your recognition of the clarity, consistency, and relevance of our work, as well as your acknowledgment of the well-presented figures and comprehensive literature basis.

In response to your suggestion, we have incorporated additional examples from the literature that highlight the potential of phosphatases as therapeutic targets in cancer therapy. We have also included relevant preclinical and clinical data to support these findings.
The updated manuscript, has been uploaded for your review.
Best regards,

Reviewer 2 Report

Comments and Suggestions for Authors

The manuscript by Fonodi et al. describes de-phosphorylation in tumour angiogenesis, focusing on the activity of PP1, PP2A and B and PTPs. The results presented are relevant to several fields, cell biochemistry, molecular biology and pharmacology.

Points to be addressed:

1) The review is comprehensive and extensive. The organisation of the paragraphs makes the review an interesting text for a general audience, however the lack of more simplified figures than those proposed by the authors makes for heavy reading. More figures illustrating the evidence in the different sections would be useful.

2) Reviews should provide a specific and critical assessment of the literature. This review is primarily a collection of information on the activities of PPs and PTPs. A list of limitations and future research needs and priorities should be included in each section. 

Author Response

Thank you for your detailed and constructive review of our manuscript. We appreciate your recognition of the relevance of our work across several fields, including cell biochemistry, molecular biology, and pharmacology.
In response to the points you have raised: We have added more simplified and illustrative figures to make the review more accessible and engaging for a general audience. These figures are designed to clearly illustrate the key concepts and evidence discussed in the different sections, thereby enhancing the readability of the manuscript. We have revised the manuscript to include a specific and critical assessment of the literature. Each section now contains a discussion of the limitations of the current research as well as future research needs and priorities. This addition aims to provide a more comprehensive and insightful review, offering valuable perspectives for further investigation in this field.
The updated manuscript, with the new figures has been uploaded for your review.